# Calibrating and Improving Graph Contrastive Learning

**Kaili Ma**[*]                                                                       *klma@cse.cuhk.edu.hk*
*Department of Computer Science and Engineering,*
*The Chinese University of Hong Kong*

**Garry Yang**[*]                                                                    *hcyang@cse.cuhk.edu.hk*
*Department of Computer Science and Engineering,*
*The Chinese University of Hong Kong*

**Han Yang**                                                                         *hyang@cse.cuhk.edu.hk*
*Department of Computer Science and Engineering,*
*The Chinese University of Hong Kong*

**Yongqiang Chen**                                                                   *yqchen@cse.cuhk.edu.hk*
*Department of Computer Science and Engineering,*
*The Chinese University of Hong Kong*

**James Cheng**                                                                      *jcheng@cse.cuhk.edu.hk*
*Department of Computer Science and Engineering,*
*The Chinese University of Hong Kong*
[*]*These authors contributed equally to this work.*

**Reviewed on OpenReview:** *https://openreview.net/forum?id=LdSP6cvTS4*

## Abstract

Graph contrastive learning algorithms have demonstrated remarkable success in various applications such as node classification, link prediction, and graph clustering. However, in unsupervised graph contrastive learning, some contrastive pairs may contradict the truths in downstream tasks and thus the decrease of losses on these pairs undesirably harms the performance in the downstream tasks. To assess the discrepancy between the prediction and the ground-truth in the downstream tasks for these contrastive pairs, we adapt expected calibration error (ECE) to graph contrastive learning. The analysis of ECE motivates us to propose a novel regularization method, **Contrast-Reg**, to ensure that decreasing the contrastive loss leads to better performance in the downstream tasks. As a plug-in regularizer, Contrast-Reg effectively improves the performance of existing graph contrastive learning algorithms. We provide both theoretical and empirical results to demonstrate the effectiveness of Contrast-Reg in enhancing the generalizability of the Graph Neural Network (GNN) model and improving the performance of graph contrastive algorithms with different similarity definitions and encoder backbones across various downstream tasks.

## 1 Introduction

Graph structures are widely used to capture abundant information, such as hierarchical configurations and community structures, present in data from various domains like social networks, e-commerce networks, knowledge graphs, the World Wide Web, and semantic webs. By incorporating graph topology along with node and edge attributes into machine learning frameworks, graph representation learning has demonstrated remarkable success in numerous essential applications, such as node classification, link prediction, and graph clustering. Graph contrastive learning has emerged as an effective method for learning graph representations, including algorithms such as DGI (Velickovic et al., 2019), GMI (Peng et al., 2020), GCC (Qiu et al., 2020)

and PyGCL (Zhu et al., 2021a). These algorithms use the noise contrastive estimation loss (NCEloss) to contrast the similarities of similar (or positive) node pairs against those of negative pairs. The algorithms differ in their definition of node similarity, hence the design of contrastive pairs. The design of the encoder backbone can vary from Graph Neural Networks (GNNs) (Kipf & Welling, 2017; Hamilton et al., 2017; Velickovic et al., 2018; Xu et al., 2019) to a skip-gram based model (Tang et al., 2015; Perozzi et al., 2014; Grover & Leskovec, 2016). In this paper, we focus on employing graph contrastive learning algorithms in an unsupervised learning manner and utilize GNNs as the encoder. The learned node embeddings are directly delivered to the downstream tasks.

Although graph contrastive algorithms have demonstrated strong performance in some downstream tasks, we have discovered that directly applying contrastive pairs and GNN models to new tasks or data is not always effective (as shown in Section 6). Nodes with different ground-truth labels can be selected as positive pairs during unsupervised training. Contrastive models that solely minimize the NCE loss can be misled by these pairs and learn certain spurious features that decrease the loss but are detrimental to downstream tasks. To address this issue, we adapt expected calibration error (ECE) to graph contrastive learning to assess the discrepancy between the predicted probability and the ground-truth in downstream tasks for the selected contrastive pairs. High ECE values indicate mis-calibration of the model. We identify two factors that contribute to the model's mis-calibration: (a) the expectation of the prediction probability for randomly sampled pairs, $\mathbb{E}(v, v')[\sigma(h_v \cdot h_{v'})]$, and (b) the probability $q^+$ of $v'_+$ sharing the same label as $v$ in positive sampling. To address the mis-calibration in existing graph contrastive learning algorithms, we introduce a novel regularization method, denoted as **Contrast-Reg**. Contrast-Reg employs a regularization vector $r$, which consists of a random vector with each entry within the range $(0, 1]$. By ensuring node representations maintain similarity with $r$ and that pseudo-negative representations, generated through shuffled node features, remain dissimilar to $r$, Contrast-Reg guarantees that minimizing the contrastive loss results in high-quality representations that improve accuracy in downstream tasks, rather than overfitting specific spurious features. We provide both theoretical evidence and empirical experiments to support the effectiveness of Contrast-Reg. First, we derive a generalization bound for our contrastive GNN framework, building upon the theoretical framework depending on Rademacher complexity (Saunshi et al., 2019), and demonstrate that Contrast-Reg contributes to a decrease in the upper bound. This result indicates that this term promotes better alignment with the performance of downstream tasks while simultaneously minimizing the training loss, thereby improving the generalizability of the GNN model. Furthermore, we design experiments to examine the empirical performance of Contrast-Reg by formulating the graph contrastive learning framework into four components: a similarity definition, a GNN encoder, a contrastive loss function, and a downstream task. We apply Contrast-Reg to different compositions of these components and achieve superior results across various compositions.

The main contributions of this paper can be summarized as follows:

- (Section 4.1) We adapt expected calibration error (ECE) to assess the discrepancy between the prediction and the ground-truth in the downstream tasks on contrastive pairs, and identify two key factors that influence the discrepancy.

- (Section 4.2) We propose a novel regularization method, Contrast-Reg, to ensure that minimizing the contrastive loss results in high-quality representations and improved accuracy in downstream tasks. Contrast-Reg is a plugin regularizer generally effective with respect to various graph contrastive learnings.

- (Section 4.2 & Section 5 & Section 6) We provide both theoretical results and empirical results to demonstrate the effectiveness of Contrast-Reg in improving the generalizability of the GNN model and achieving superior performance across different existing graph contrastive learning algorithms.

## 2   Related Work

**Graph representation learning.** Many graph representation learning models have been proposed. Factorization-based models (Ahmed et al., 2013; Cao et al., 2015; Qiu et al., 2018) factorize an adjacency matrix to obtain node representations. Random walk-based models such as DeepWalk (Perozzi et al., 2014)

sample node sequences as the input to skip-gram models to compute the representation for each node. Node2vec (Grover & Leskovec, 2016) balances depth-first and breadth-first random walk when it samples node sequences. HARP (Chen et al., 2018) compresses nodes into super-nodes to obtain a hierarchical graph to provide hierarchical information to random walk. GNN models (Kipf & Welling, 2017; Hamilton et al., 2017; Velickovic et al., 2018; Xu et al., 2019; Qu et al., 2019; Thomas et al., 2023) have shown great capability in capturing both graph topology and node/edge feature information. Most GNN models follow a neighborhood aggregation schema, in which each node receives and aggregates the information from its neighbors in each GNN layer, i.e., for the $k$-th layer, $\tilde{h}_i^k = aggregate(h_j^{k-1}, j \in neighborhood(i))$, and $h_i^k = combine(\tilde{h}_i^k, h_i^{k-1})$. This work employs GNN models as the backbone and tests the representation across various downstream tasks, such as node classification, link prediction, and graph clustering.

**Graph contrastive learning.** Contrastive learning is a self-supervised learning method that learns representations by contrasting positive pairs against negative pairs. Contrastive pairs can be constructed in various ways for different types of data and tasks, such as multi-view (Tian et al., 2020; 2019), target-to-noise (van den Oord et al., 2018; Hénaff et al., 2019), mutual information (Belghazi et al., 2018; Hjelm et al., 2019), instance discrimination (Wu et al., 2018), context co-occurrence (Mikolov et al., 2013), clustering (Asano et al., 2020; Caron et al., 2018), multiple data augmentation (Chen et al., 2020), known and novel pairs (Sun & Li, 2023), and contextually relevant (Neelakantan et al., 2022). Contrastive learning has been successfully applied to numerous graph representation learning models (Velickovic et al., 2019; Peng et al., 2020; You et al., 2020; Qiu et al., 2020; Zeng et al., 2021) to pseudo subgraph instance discrimination as a contrastive learning training objective and to leverage contrastive learning to empower graph neural networks in learning node representations. We characterize different types of node-level similarity as follows:

- **Structural similarity**: Structural similarity can be captured from various perspectives. From a graph theory viewpoint, GraphWave (Donnat et al., 2018) leverages the diffusion of spectral graph wavelets to capture structural similarity, while struc2vec (Ribeiro et al., 2017) uses a hierarchy to measure node similarity at different scales. From an induced subgraph perspective, GCC (Qiu et al., 2020) treats the induced subgraphs of the same ego network as similar pairs and those from different ego networks as dissimilar pairs. To capture community structure, vGraph (Sun et al., 2019) utilizes the high correlation between community detection and node representations to incorporate more community structure information into node representations. To capture global-local structure, DGI (Velickovic et al., 2019) maximizes the mutual information between node representations and graph representations to allow node representations to contain more global information. GDCL (Zhao et al., 2021) incorporates the results of graph clustering to decrease the false-negative samples. DiGCL (Tong et al., 2021) generates contrastive pairs by Laplacian perturbations to retain more structural features of directed graphs. GCA (Zhu et al., 2021b) constructs contrastive pairs by designing adaptive augmentation schemes based on node centrality measures. AD-GCL (Suresh et al., 2021) adopts trainable edge-dropping graph augmentation and optimizes both adversarial graph augmentation strategies and contrastive objectives to avoid the model learning redundant graph features.

- **Attribute similarity**: Nodes with similar attributes are likely to have similar representations. GMI (Peng et al., 2020) maximizes the mutual information between node attributes and high-level representations, and Pretrain GNN (Hu et al., 2020b) applies attribute masking to help capture domain-specific knowledge. GCA (Zhu et al., 2021b) corrupts node features by adding more noise to unimportant node features, to enforce the model to recognize underlying semantic information.

Given the above subgraph instance discrimination objective with GNN backbones, NCEloss (Gutmann & Hyvärinen, 2012; Dyer, 2014; Mnih & Teh, 2012; Yang et al., 2020) is applied to optimize the model's parameters. In addition, Graph Autoencoder, which is another popular graph self-supervised approach by reconstructing useful graph information, has also been extensively studied. For example, GraphMAE (Hou et al., 2022) focuses on feature reconstruction with a masking strategy and scaled cosine error. GraphMAE2 (Hou et al., 2023) designs strategies of multi-view random re-mask decoding and latent representation prediction to regularize the reconstruction process. S2GAE (Tan et al., 2023) randomly masks a portion of edges and

learns to reconstruct these missing edges with a novel masking strategy. SeeGera (Li et al., 2023) adopts a novel hierarchical variational framework.

Numerous studies have demonstrated the success of enhancing a model's generalizability by devising advanced similarity functions. However, some designs may require extensive tuning, preventing these advanced techniques from adapting to settings beyond the scope of their experiments. This problem is especially prevalent in the unsupervised learning setting, where the ideal protocol is that the learned node embeddings, which are the output of the unsupervised contrastive model, should be applicable to downstream tasks upon convergence. The key difference between Contrast-Reg and other algorithms, such as GCA, GDCL, and DiGCL, lies in the fact that the latter methods are intended to incorporate structures that are critical to the performance of the downstream tasks, while Contrast-Reg ensures that minimizing the contrastive loss results in high-quality representations that enhance the accuracy in downstream tasks. Moreover, Contrast-Reg can naturally function as a plugin to these advanced graph contrastive learning algorithms to improve their performance.

**Expected Calibration Error.** Expected Calibration Error (ECE) (Guo et al., 2017) is a metric employed to quantify the calibration between *confidence* (largest predicted probability) and *accuracy* in a model. Calibration refers to the consistency between a model's predicted probabilities and the actual outcomes. When a model is mis-calibrated, its generalization to unseen data in supervised learning tasks is likely to be poor (Müller et al., 2019; Pereyra et al., 2017; Guo et al., 2017; Zhang et al., 2018). We propose using ECE to evaluate the quality of node embeddings generated by unsupervised graph contrastive learning models. This calibration offers insights into addressing the challenge that applying graph contrastive learning algorithms to downstream tasks does not always yield optimal results.

**Regularization for graph representation learning.** GraphAT (Feng et al., 2019) and BVAT (Deng et al., 2019) introduce adversarial perturbations $\frac{\partial f}{\partial x}$ to the input data $x$ as regularizers to obtain more robust models. GraphSGAN (Ding et al., 2018) generates fake input data in low-density regions by incorporating a generative adversarial network as a regularizer. P-reg (Yang et al., 2021) leverages the smoothness property in real-world graphs to enhance GNN models. Graphnorm (Cai et al., 2021) proposes a novel feature normalization method. ALS (Zhou et al., 2021) employs label propagation to adaptively integrate label smoothing into GNN training. G-Mixup (Han et al., 2022) uses graphons as a surrogate to apply mixup techniques to graph data.

The above regularizers are designed for general representation generalizability, while Contrast-Reg is specifically intended to address the mis-calibration problem in the unsupervised graph contrastive learning optimization process and its performance on downstream tasks. It is worth noting that Contrast-Reg could be used in conjunction with the aforementioned regularization techniques.

## 3 Preliminaries

We begin by introducing the concepts and foundations of graph contrastive learning (GCL). Let a graph $\mathcal{G} = (\mathcal{V}, \mathcal{E})$ be denoted, where $\mathcal{V} = v_1, v_2, \cdots, v_n$ and $\mathcal{E}$ represent the vertex set and edge set of $\mathcal{G}$, and the node feature vector of node $v_i$ be $x_i$. Our objective is to learn node embeddings through unsupervised graph contrastive learning and subsequently apply simple classifiers leveraging these embeddings for various downstream tasks, such as node classification, link prediction, and graph clustering.

**Graph contrastive learning** Given a graph $\mathcal{G}$ with node features $\mathcal{X} = (x_1, x_2, \cdots, x_n)$, the aim of graph contrastive learning is to train an encoder $f : (\mathcal{G}, \mathcal{X}) \rightarrow \mathbb{R}^d$ for all input data points $v_i \in \mathcal{V}$ with node feature vector $x_i$ by constructing positive pairs $(v_i, v_i^+)$ and negative pairs $(v_i, v_{i1}^-, \cdots, v_{iK}^-)$.

The encoder $f$ is typically implemented using Graph Neural Networks (GNNs). Specifically, let $h_i$ represent the output embedding of the encoder $f$ for node $v_i$, $h_i^{(k)}$ as the embedding at the $k$-th layer, and $h_i^0 = x_i$.

The output of the encoder $f$ is then iteratively defined as:

$$m_i^{(k)} = \text{Aggregate}_k \left( \left\{ h_j^{(k-1)} : v_j \in \mathcal{N}(v_i) \right\} \right)$$
$$h_i^{(k)} = \text{RELU} \left( W^k \cdot \text{Update} \left( h_i^{(k-1)}, m_i^{(k)} \right) \right), \tag{1}$$

where $\mathcal{N}(v_i)$ the set of nodes adjacent to $v_i$, and Aggregate and Update are the aggregation and update function of GNNs (Gilmer et al., 2017). $h_i$ is the last layer of $h_i^{(k)}$ for node $v_i$.

Leveraging various types of similarity as pseudo subgraph instance discrimination labels, graph contrastive learning constructs positive and negative pairs to train the embedding $h_i$ by optimizing the loss on these pairs. The most commonly employed loss is the NCEloss:

$$\hat{\mathcal{L}}_{nce} = \frac{1}{M} \sum_{i=1}^{M} \left[ -\log \sigma(h_i^T h_i^+) + \sum_{k=1}^{K} \log \sigma(h_i^T h_{ij}^-) \right] \tag{2}$$

with $M$ samples $\left( v_i, v_i^+, v_{i1}^-, \cdots, v_{iK}^- \right)_{i=1}^{M}$ in empirical setting where $\sigma(\cdot)$ is the sigmoid function.

**Calibrating graph contrastive learning by the expected calibration error (ECE)**  Expected Calibration error measures the degree to which the model output probabilities match ground-truth accuracy in supervised tasks (Naeini et al., 2015; Guo et al., 2017). It's defined as the expectation of absolute difference between the *largest predicted probability (confidence)* and its corresponding *accuracy*,

$$\text{ECE} = \mathbb{E}_{(v,v') \in S} \left[ |p(v, v') - acc(v, v')| \right] \tag{3}$$

In this paper, we extend the use of ECE to evaluate the quality of graph contrastive learning. Specifically, we can compare the predicted probability of a positive pair (i.e., a pair of nodes with the same label) with the true probability that the pair belongs to the same class. We can also compare the predicted probability of a negative pair (i.e., a pair of nodes with different labels) with the true probability that the pair belongs to different classes. By calculating the differences between the predicted and true probabilities for both positive and negative pairs, we can quantify the overall mis-calibration of the model and identify areas for improvement. The formal definition is presented as follows. The largest predicted probability, $p(v, v')$, is determined as

$$p(v, v') = \begin{cases} \sigma(h_{v'} \cdot h_v), & (v, v') \text{ as positive pair} \\ 1 - \sigma(h_{v'} \cdot h_v), & (v, v') \text{ as negative pair} \end{cases} \tag{4}$$

where $h_v$ and $h_{v'}$ are the embeddings for the target node $v$ and the selected sample $v'$, respectively, and $\sigma(\cdot)$ is the sigmoid function. The corresponding accuracy, $acc(v, v')$, is defined as follows:

$$acc(v, v') = \begin{cases} \mathbb{I}(v, v'), & (v, v') \text{ as positive pair} \\ 1 - \mathbb{I}(v, v'), & (v, v') \text{ as negative pair} \end{cases} \tag{5}$$

where $\mathbb{I}$ is an indicator function denoting whether $v$ and $v'$ has the same label, the positive/negative pairs are pseudo-labels for training the contrastive learning algorithm which may not equal to the true label. In this case, when two nodes are selected as positive pairs, $acc(v, v')$ is equal to 1 if $v'$ and $v$ belong to the same class and equal to 0 otherwise; when the two nodes are selected as negative pairs, $acc(v, v') = 1 - \mathbb{I}(v, v')$.

Using the ECE metric in this way allows us to evaluate the quality of the embeddings outputted by the model ($\sigma(h_{v'} \cdot h_v)$) and their accuracy in downstream tasks ($acc(v, v')$), such as node classification. We can improve the general performance and utility of graph contrastive learning algorithms by detecting regions of mis-calibration and developing novel techniques for enhancing calibration performance.

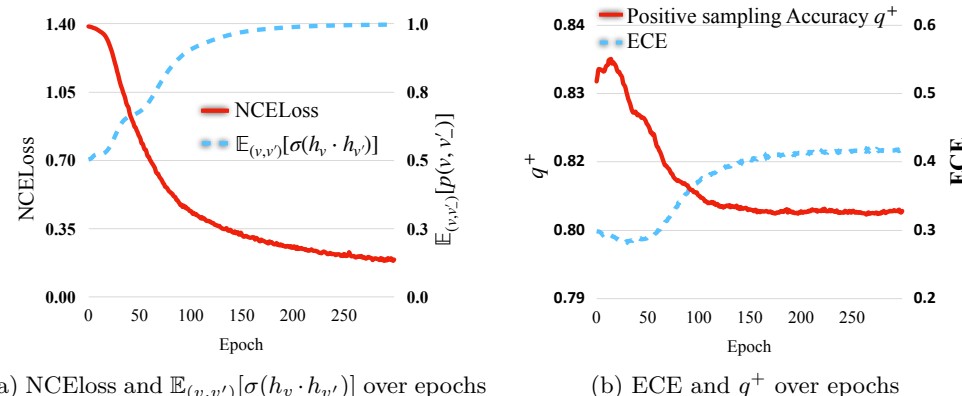

(a) NCEloss and $\mathbb{E}_{(v,v')}[\sigma(h_v \cdot h_{v'})]$ over epochs

(b) ECE and $q^+$ over epochs

Figure 1: Calibrating Graph Contrastive Learning (LC (Alg. 3) w/o Contrast-Reg on the Pubmed dataset)

## 4 Methodology

To calibrate the model, we first adapt the expected calibration error (ECE) formula for the graph contrastive learning setting (Section 4.1). In order to ensure that minimizing the contrastive loss leads to high-quality representations that improve accuracy in downstream tasks, we propose a regularization term designed to lower the ECE value and enhance the model's generalizability (Section 4.2). We also provide theoretical evidence to support the effectiveness of the proposed regularization term.

### 4.1 Empirical Calibration Reveals Limitations of Existing Graph Contrastive Learning Algorithms

To calibrate the existing graph contrastive learning algorithms, we measure the degree of calibration using the expected calibration error (ECE) metric, which is defined by Equation 3, 4, 5:

$$
\begin{aligned}
\text{ECE} = r^+ \cdot \Bigg( & \left(1 - \mathbb{E}_{acc(v,v'_+)=1}[p(v,v'_+)]\right) q^+ && \textit{(true positive)} \\
& + \mathbb{E}_{acc(v,v'_+)=0}[p(v,v'_+)] \cdot \left(1 - q^+\right) \Bigg) && \textit{(false positive)} \\
+ r^- \Bigg( & \mathbb{E}_{acc(v,v'_-)=1}[p(v,v'_-)] \cdot q^- && \textit{(false negative)} \\
& + \left(1 - \mathbb{E}_{acc(v,v'_-)=0}[p(v,v'_-)]\right) \cdot (1 - q^-) \Bigg) && \textit{(true negative)},
\end{aligned}
\tag{6}
$$

where $q^+$ and $q^-$ are the probabilities that the node $v'$ has the same label as node $v$ in positive and negative sampling, respectively; $r^+$ and $r^-$ are the ratios of sampling positive and negative samples, with $r^+ + r^- = 1$. In this study, we set $r^+ = r^- = 0.5$.

Building upon this formulation, we propose the following claim that takes into account the positive and negative pair construction assumptions to analyze the potential issues that lead to the mis-calibration between the decrease in graph contrastive learning training loss and the degradation of downstream task performance.

**Claim 4.1.** *Under the assumption that negative sampling is uniformly sampled, and positive sampling is sampled based on the calculated distance between pairwise embeddings, ECE is positively correlated with the expectation of the prediction value for randomly sampled pairs $\mathbb{E}_{v,v'}[\sigma(h_v \cdot h_{v'})]$, and negatively correlated with the probability $q^+$ of $v'_+$ having the same label as $v$ in positive sampling.*

We provide a thorough analysis of this claim in Appendix A.1. To investigate the changes in these two factors and the ECE value over the process of minimizing the NCEloss (Equation 2), we conduct an illustrative

experiment with an existing graph contrastive learning algorithm and present the relevant values in Figure 1. In Figure 1a, we show that as the NCEloss (red solid line) is minimized over the epochs, the expectation of the prediction value for randomly sampled pairs $\mathbb{E}(v, v')[\sigma(h_v \cdot hv')]$ (blue dashed line) increases. Moreover, Figure 1b demonstrates the model's challenge in identifying genuine positive pairs, with the probability $q^+$ initially increasing before gradually declining (red solid line), and the ECE value first decreasing before rising (blue dashed line). These findings suggest that, as the epochs progress, merely reducing the NCEloss is insufficient to enhance the accuracy of downstream tasks. Our empirical calibration shows that the model learns certain spurious features that reduce the loss but are ultimately harmful to downstream tasks when solely minimizing the NCEloss. The above analysis emphasizes the need for graph contrastive learning algorithms to effectively quantify the relationships between learned embeddings and the accuracy of downstream tasks. In the next section, we present our approach to mitigate the possible risks associated with spurious feature learning in graph contrastive learning algorithms and show its effectiveness through a number of experiments.

### 4.2 Proposed Regularization Term: Contrast-Reg

To ensure that minimizing the NCEloss aligns with downstream task accuracy, we propose a contrastive regularization term, denoted as **Contrast-Reg**, given by:

$$\mathcal{L}_{reg} = - \mathop{\mathbb{E}}_{h,\tilde{h}} \left[ \log \sigma(h_i^T W \mathbf{r}) + \log \sigma(-\tilde{h}_i^T W \mathbf{r}) \right], \tag{7}$$

where $\mathbf{r}$ is a random vector uniformly sampled from $(0, 1]$, $W$ is a trainable parameter, and $\tilde{h}$ is the noisy feature generated by different data augmentation techniques, such as those used in the previous literature (Chen et al., 2020; Velickovic et al., 2019). In Section 5, we will discuss how we calculate the noisy features in the GNN setting. In the following, we will introduce the impact of Contrast-Reg on the ECE value and the generalizability of the GNN model.

**ECE decreases by incorporating Contrast-Reg**   In Claim 4.1 and Appendix A.1, it is proven that the ECE value is positively correlated with $\mathbb{E}(v, v')[\sigma(h_v, h_{v'})]$ and inversely correlated with $q^+$. In addition, Theorem 2 in Appendix A.3 shows that Contrast-Reg reduces the variance of the learned node embeddings' norm $\|h\|$, thereby preventing the loss from decreasing due to exploding embedding norms. As a result, the norm values are lower than those obtained with the vanilla contrastive loss, leading to a reduction in $\mathbb{E}(v, v')[\sigma(h_v, h_{v'})]$ and an increase in representation quality. With the improvement in the representation quality, $\mathbb{E}(v, v')[\sigma(h_v, h_{v'})]$ decreases and $q^+$ increases, resulting in a decreased ECE value. Furthermore, we conducted an empirical study to examine the impact of Contrast-Reg on the ECE value and investigate the two factors that cause miscalibration. Figure 2a shows that the expectation of the prediction value for randomly sampled pairs $\mathbb{E}_{(v,v')}[\sigma(h_v \cdot h_{v'})]$ with Contrast-Reg increased much more slowly compared to the vanilla NCEloss (green solid line compared to the red solid line). Figure 2b presents the changes in positive sampling accuracy $q^+$ over the epochs of the vanilla NCEloss and NCEloss with Contrast-Reg, where Contrast-Reg helps $q^+$ increase, while $q^+$ trained by vanilla NCEloss decreased after the initial increases. These two factors contribute to the different ECE changes in the vanilla NCEloss and NCEloss with Contrast-Reg. The red dashed line indicates that the ECE value increases after the initial decreases, while the green dashed line shows that the ECE value decreases slightly. The comparison demonstrates that applying Contrast-Reg alleviates the miscalibration in representation learning, ensuring that minimizing the contrastive loss results in high-quality representations with increased accuracy in downstream tasks, rather than overfitting certain spurious features. We provide a detailed comparison of the impact on the ECE value between models with and without Contrast-Reg in Appendix A.2 across different datasets.

**Generalizability improved by incorporating Contrast-Reg**   In Section 6.3, we present empirical evidence supporting the effectiveness of Contrast-Reg's ability to enhance the generalizability of the GNN model through an ablation study. Furthermore, we analyze the impact of Contrast-Reg on the generalizability of graph contrastive learning algorithms using the Rademacher complexity (Saunshi et al., 2019). In Theorem 1, we offer performance guarantees for the learned graph embeddings outputted by the GNN function class $\mathcal{F} = f$ with the unsupervised loss function $\mathcal{L}_{nce}$ on the downstream average classification task $\mathcal{L}_{sup}^\mu(\hat{f})$. Detailed settings can be found in Appendix A.4. Assume that $f$ is bounded, i.e., $\|h_i\| \leq R$

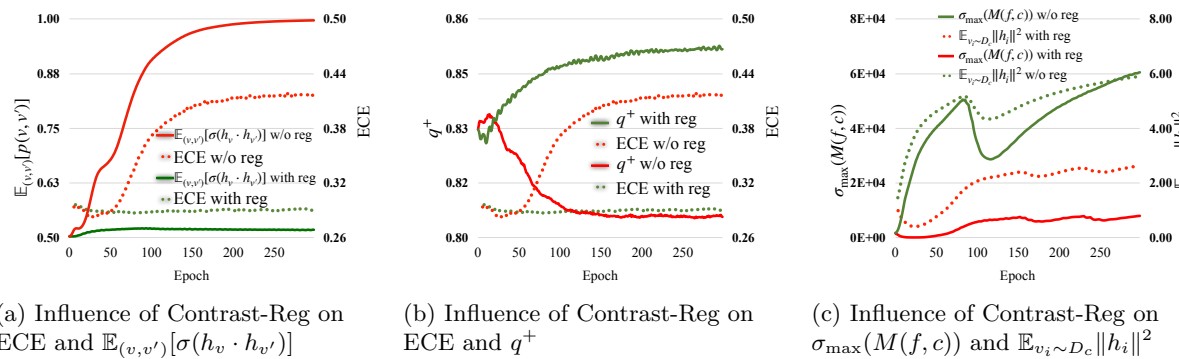

(a) Influence of Contrast-Reg on ECE and $\mathbb{E}_{(v,v')}[\sigma(h_v \cdot h_{v'})]$

(b) Influence of Contrast-Reg on ECE and $q^+$

(c) Influence of Contrast-Reg on $\sigma_{\max}(M(f,c))$ and $\mathbb{E}_{v_i \sim D_c}\|h_i\|^2$

Figure 2: Effects of Contrast-Reg (LC (Alg. 3) w/ & w/o Contrast-Reg on Pubmed)

with $R > 0$. Let $c, c'$ be two classes sampled independently from latent classes $\mathcal{C}$ with distribution $\rho$. Let $\tau = \mathbb{E}_{c,c' \sim \rho^2} \mathbb{I}(c = c')$ be the probability that $c$ and $c'$ come from the same class. Let $\mathcal{L}_{nce}^{\neq}(f)$ be the NCEloss when negative samples come from different classes. We have the following theorem.

**Theorem 1.** $\forall f \in \mathcal{F}$, with probability at least $1 - \delta$,

$$\mathcal{L}_{sup}^{\mu}(\hat{f}) \leq \mathcal{L}_{nce}^{\neq}(f) + \beta s(f) + \eta Gen_M, \tag{8}$$

where $Gen_M = \frac{8 R \mathcal{R}_\mathcal{S}(\mathcal{F})}{M} - 8 \log(\sigma(-R^2)) \sqrt{\frac{\log \frac{4}{\delta}}{2M}} = O\left(R \frac{\mathcal{R}_\mathcal{S}(\mathcal{F})}{M} + R^2 \sqrt{\frac{\log \frac{1}{\delta}}{M}}\right)$, $\beta = \frac{\tau}{1-\tau}$, $\eta = \frac{1}{1-\tau}$, and $s(f) = 4\sqrt{\mathbb{E}_{(v_i,v_j) \sim \mathcal{D}_{sim}(v_i,v_j)}[(h_i^T h_i)^2]}$,

In Equation (8), $Gen_M$ represents the generalization error in terms of *Rademacher complexity* and will converge to 0 when the encoder function $f$ is bounded and the number of samples $M$ is sufficiently large. The theorem above indicates that only when $\beta s(f) + \eta Gen_M$ converges to 0 as $M$ increases, will the encoder $\hat{f} = \arg\min_{f \in \mathcal{F}} \hat{\mathcal{L}}_{nce}$ selected perform well in downstream tasks.

Next, we will analyze the impact of Contrast-Reg on the condition in the aforementioned theorem, $\beta s(f) + \eta Gen_M$, by reformulating $s(f)$ as follows:

$$\begin{aligned} s(f) &= 4\sqrt{\mathbb{E}_{(v_i,v_j) \sim \mathcal{D}_{sim}(v_i,v_j)}\left[h_i^T h_j h_j^T h_i\right]} \\ &= 4\sqrt{\mathbb{E}_{c \sim \rho}\left[\mathbb{E}_{v_i \sim \mathcal{D}_c}\left[h_i^T \mathbb{E}_{x_j \sim \mathcal{D}_c}\left[h_j h_j^T\right] h_i\right]\right]} \\ &\leq 4\sqrt{\mathbb{E}_{c \sim \rho}\left[\|M(f,c)\|_2 \mathbb{E}_{v_i \sim \mathcal{D}_c}\|h_i\|^2\right]}, \end{aligned}$$

where $M(f,c) := \mathbb{E}_{v_i \sim \mathcal{D}_c}\left[h_i h_i^T\right]$.

The advantage of incorporating Contrast-Reg into graph contrastive learning lies in its ability to reduce both the largest singular value of matrix $M(f,c)$ ($\sigma_{\max}(M(f,c))$) and the expectation of the embedding norm within the same class, $\mathbb{E}_{v_i \sim D_c}[\|h_i\|^2]$ (as illustrated in Figure 2c). This reduction leads to a decrease in the upper bound of $s(f)$, ultimately yielding a lower value for the term $\beta s(f) + \eta Gen_M$ compared to vanilla graph contrastive learning. The reduction in this term promotes better alignment with the performance of downstream tasks while simultaneously minimizing the training loss. Further experiments on the generalizability of the Contrast-Reg approach can be found in Section 6.3, and a more detailed explanation on lowering the term $s(f)$ is provided in Appendix A.3.

## 5 A Contrastive GNN Framework

Algorithm 1 presents the graph contrastive learning framework and how Contrast-Reg is plugged into the framework. The framework employs a GNN model, such as GCN (Kipf & Welling, 2017), GAT (Velickovic

et al., 2018), and GIN (Xu et al., 2019), to obtain node embeddings. To train the parameters of the GNN backbone, we generate contrastive pairs using the functions *SeedSelect* and *Contrast*. In addition, we incorporate the Contrast-Reg term into the optimization process, along with the contrastive loss on these pairs. After the model convergence, the embeddings are directly delivered to the downstream tasks.

---

**Algorithm 1:** Graph Contrastive Learning Framework

---

**Input:** Graph $\mathcal{G} = (\mathcal{V}, \mathcal{E})$, node attributes $\mathcal{X}$, a GNN model $f : \mathcal{V} \to \mathbb{R}^H$, the number of epochs $e$;
**Output:** A trained GNN model $f$;

**1** Initialize training parameters;
**2** **for** *epoch* $\leftarrow 1$ **to** $e$ **do**
**3**     $\mathcal{C}$=SeedSelect($\mathcal{G}$, $\mathcal{X}$, $f$, *epoch*);
**4**     $\mathcal{P}$=Contrast($\mathcal{C}$, $\mathcal{G}$, $\mathcal{X}$, $f$);
**5**     loss = NCEloss($\mathcal{P}$) + Contrast-Reg($\mathcal{G}$, $\mathcal{X}$, $f$);
**6**     Back-propagation and update $f$;
**7** **end**

---

The functions *SeedSelect* and *Contrast* are responsible for selecting the seed nodes and sampling the positive/negative pairs for these seeds while incorporating various priors for both structural and attribute aspects of the graph. To illustrate the graph contrastive learning framework, we utilize two simple graph contrastive learning algorithms (**ML** and **LC**).

- **LC: Structural similarity** We adopt clustering methods to detect communities in a graph and generate positive/negative pairs in the node representation space (Newman, 2006). The *SeedSelect* function is implemented using curriculum learning, following the design of AND (Huang et al., 2019). Initially, *SeedSelect* utilizes node embeddings to select a node set $\mathcal{C}$ with low entropy, reducing randomness and uncertainty during early training stages. As training progresses, *SeedSelect* gradually expands $\mathcal{C}$ by adding more nodes. For each node $x_i$ in $\mathcal{C}$, *Contrast* generates a positive node $x_i^+$ that is highly similar, and a randomly sampled negative node $x_i^-$, returning a tuple of their representations: $\left\{ (f(x_i), f(x_i^+), f(x_i^-)) \right\}_{x_i \in \mathcal{C}}$.

- **ML: Attribute similarity** In ML, we assume that nodes with similar attributes should have similar representations to preserve the attribute information. Following the design of DIM (Hjelm et al., 2019) and GMI (Peng et al., 2020), we adopt a multi-level representation approach. Initially, *SeedSelect* includes all nodes in the node set $\mathcal{G}$. For each seed node $x_i$, *Contrast* uses the node itself as the positive node and randomly samples a negative node. The positive representation of the seed node $x_i$ is generated by adding an additional GNN layer upon $f$, denoted as $g$. The tuple containing the representations is denoted as $\left\{ (f(x_i), g(x_i), f(x_i^-)) \right\}_{x_i \in \mathcal{G}}$.

Advanced graph contrastive learning algorithms can also be easily implemented using this framework. As an example, Graph Contrastive Analysis (GCA) (Zhu et al., 2021b) algorithm can be implemented by making a simple modification to the *Contrast* function found in Appendix B. Overall, Contrast-Reg serves as a highly effective plugin regularizer for graph contrastive learning with various backbones, and with different *Contrast* and *SeedSelect* implementations.

## 6 Experimental Results

We begin by introducing the experimental settings in Section 6.1. Section 6.2 presents the main results across various downstream tasks. Moreover, we assess the benefits of Contrast-Reg through ablation studies.

### 6.1 Experiment Settings

**Downstream tasks** We conduct our experiments on three distinct downstream tasks, namely node classification, graph clustering, and link prediction. The experimental procedure consists of two stages: first, we utilize positive and negative contrastive pairs to train the GNN models in an unsupervised manner, obtaining the node embeddings. Subsequently, we apply these embeddings to the downstream tasks by integrating

Table 1: Downstream task: node classification

| Algorithm | Contrast-Reg | Cora | Citeseer | Pubmed | Wiki | Computers | Photo | ogbn-arxiv | ogbn-products | Reddit |
|---|---|---|---|---|---|---|---|---|---|---|
| GCN | | $81.54_{\pm0.68}$ | $71.25_{\pm0.67}$ | $79.26_{\pm0.38}$ | $72.40_{\pm0.95}$ | $79.82_{\pm2.04}$ | $88.75_{\pm1.99}$ | $\mathbf{71.74}_{\pm0.29}$ | $75.64_{\pm0.21}$ | $94.02_{\pm0.05}$ |
| node2vec | | $71.07_{\pm0.91}$ | $47.37_{\pm0.95}$ | $66.34_{\pm1.40}$ | $58.76_{\pm1.48}$ | $75.37_{\pm1.52}$ | $83.63_{\pm1.53}$ | $70.07_{\pm0.13}$ | $72.49_{\pm0.10}$ | $93.26_{\pm0.04}$ |
| DGI | | $81.90_{\pm0.84}$ | $71.85_{\pm0.37}$ | $76.89_{\pm0.53}$ | $63.70_{\pm1.43}$ | $64.92_{\pm1.93}$ | $77.19_{\pm2.60}$ | $69.66_{\pm0.18}$ | $\mathbf{77.00}_{\pm0.21}$ | $94.14_{\pm0.03}$ |
| GMI | | $80.95_{\pm0.65}$ | $71.11_{\pm0.15}$ | $77.97_{\pm1.04}$ | $63.35_{\pm1.03}$ | $79.27_{\pm1.64}$ | $87.08_{\pm1.23}$ | $68.36_{\pm0.19}$ | $75.55_{\pm0.39}$ | $94.19_{\pm0.04}$ |
| GCA | | $82.51_{\pm0.75}$ | $70.83_{\pm0.89}$ | $85.69._{\pm1.03}$ | $76.38_{\pm1.25}$ | $87.47_{\pm0.49}$ | $92.07_{\pm1.0}$ | OOM | OOM | OOM |
| GraphMAE | | $84.14_{\pm0.50}$ | $73.06_{\pm0.34}$ | $80.98_{\pm0.41}$ | $77.40_{\pm0.18}$ | $80.77_{\pm1.43}$ | $88.93_{\pm1.58}$ | $71.47_{\pm0.27}$ | OOM | OOM |
| GCA | ✓ | $83.90_{\pm1.22}$ | $72.39_{\pm0.88}$ | $\mathbf{87.06}_{\pm0.26}$ | $77.71_{\pm0.06}$ | $\mathbf{88.21}_{\pm0.37}$ | $\mathbf{93.13}_{\pm0.32}$ | OOM | OOM | OOM |
| GraphMAE | ✓ | $\mathbf{84.18}_{\pm0.49}$ | $\mathbf{73.52}_{\pm0.32}$ | $81.43_{\pm0.41}$ | $\mathbf{78.22}_{\pm0.13}$ | $81.16_{\pm1.10}$ | $90.71_{\pm1.29}$ | $\mathbf{71.56}_{\pm0.19}$ | OOM | OOM |
| LC | ✓ | $82.33_{\pm0.41}$ | $72.88_{\pm0.39}$ | $79.33_{\pm0.59}$ | $69.19_{\pm1.13}$ | $81.98_{\pm1.52}$ | $87.59_{\pm1.50}$ | $69.94_{\pm0.11}$ | $\mathbf{76.96}_{\pm0.34}$ | $\mathbf{94.43}_{\pm0.03}$ |
| ML | ✓ | $82.65_{\pm0.57}$ | $72.98_{\pm0.41}$ | $80.10_{\pm1.04}$ | $67.20_{\pm0.96}$ | $82.11_{\pm1.47}$ | $86.78_{\pm1.70}$ | $70.05_{\pm0.09}$ | $76.27_{\pm0.20}$ | $94.38_{\pm0.04}$ |

Table 2: Downstream task: graph clustering

| Algorithm | Contrast-Reg | Cora | | | Citeseer | | | Wiki | | |
|---|---|---|---|---|---|---|---|---|---|---|
| | | Acc | NMI | F1 | Acc | NMI | F1 | Acc | NMI | F1 |
| node2vec | | $61.78_{\pm0.30}$ | $44.47_{\pm0.21}$ | $62.65_{\pm0.26}$ | $39.58_{\pm0.37}$ | $24.23_{\pm0.27}$ | $37.54_{\pm0.39}$ | $43.29_{\pm0.58}$ | $37.39_{\pm0.52}$ | $36.35_{\pm0.51}$ |
| DGI | | $\mathbf{71.81}_{\pm1.01}$ | $54.90_{\pm0.66}$ | $\mathbf{69.88}_{\pm0.90}$ | $68.60_{\pm0.47}$ | $43.75_{\pm0.50}$ | $64.64_{\pm0.41}$ | $44.37_{\pm0.92}$ | $42.20_{\pm0.90}$ | $40.16_{\pm0.72}$ |
| AGC | | $68.93_{\pm0.02}$ | $53.72_{\pm0.04}$ | $65.62_{\pm0.01}$ | $68.37_{\pm0.02}$ | $42.44_{\pm0.03}$ | $63.73_{\pm0.02}$ | $49.54_{\pm0.07}$ | $47.02_{\pm0.09}$ | $42.16_{\pm0.11}$ |
| GMI | | $63.44_{\pm3.18}$ | $50.33_{\pm1.48}$ | $62.21_{\pm3.46}$ | $63.75_{\pm1.05}$ | $38.14_{\pm0.84}$ | $60.23_{\pm0.79}$ | $42.81_{\pm0.40}$ | $41.53_{\pm0.20}$ | $38.52_{\pm0.22}$ |
| LC | ✓ | $70.04_{\pm2.04}$ | $55.08_{\pm0.75}$ | $67.36_{\pm2.17}$ | $67.90_{\pm0.74}$ | $43.63_{\pm0.57}$ | $64.21_{\pm0.60}$ | $50.12_{\pm0.96}$ | $49.70_{\pm0.49}$ | $43.74_{\pm0.97}$ |
| ML | ✓ | $71.59_{\pm1.07}$ | $\mathbf{56.01}_{\pm0.64}$ | $68.11_{\pm1.32}$ | $\mathbf{69.17}_{\pm0.43}$ | $\mathbf{44.47}_{\pm0.46}$ | $\mathbf{64.74}_{\pm0.41}$ | $\mathbf{53.13}_{\pm1.01}$ | $\mathbf{51.81}_{\pm0.57}$ | $\mathbf{46.11}_{\pm0.93}$ |

additional straightforward models. For example, we employ multiclass logistic regression for the node classification, $k$-means for graph clustering, and a single MLP layer for link prediction. Moreover, we conduct experiments in the pretrain-finetune paradigm, as this approach constitutes a significant component of graph contrastive learning (Qiu et al., 2020). We first employ a large graph to train the GNN models, followed by finetuning such models and train a simple downstream classifier on a separate graph.

**Training details** The datasets we employ encompass citation networks, web graphs, co-purchase networks, and social networks. Comprehensive statistics for these datasets can be found in Appendix D. For Cora, Citeseer, Pubmed, ogbn-arxiv, ogbn-products, and Reddit, we adhere to the standard dataset splits and conduct 10 different runs with fixed random seeds ranging from 0 to 9. For Computers, Photo, and Wiki, we randomly divide the train/validation/test sets, allocating 20/30/all remaining nodes per class, in accordance with the recommendations in the previous literature (Shchur et al., 2018). We measure performance across 25 (5×5) different runs, comprising 5 random splits and 5 fixed-seed runs (from 0 to 4) for each random split. The hyperparameter configurations can be found in Appendix D.

## 6.2 Main Results

Our primary results involve comparing our proposed Contrast-Reg, along with the chosen similarity definitions, against state-of-the-art algorithms across various downstream tasks.

### 6.2.1 Node Classification

We evaluate node classification performance on all datasets, utilizing both full-batch training and stochastic mini-batch training. Our methods are compared with DGI (Velickovic et al., 2019), GMI (Peng et al., 2020), node2vec (Grover & Leskovec, 2016), GCA (Zhu et al., 2021b), GraphMAE (Hou et al., 2022) and supervised GCN (Kipf & Welling, 2017). GCA, DGI, and GMI represent state-of-the-art algorithms in unsupervised graph contrastive learning. Node2vec is an exemplary algorithm for random walk-based graph representation algorithms (Grover & Leskovec, 2016; Tang et al., 2015; Perozzi et al., 2014), while GCN is a classic semi-supervised GNN model. We incorporate the Contrast-Reg into ML, LC, GCA and GraphMAE models and compare them against the baseline models. The results, presented in Table 1, demonstrate that GCA and GraphMAE with Contrast-Reg achieve excellent performance across all the datasets that are trained in the full-batch mode and even surpass the performance of the supervised GCN.

Table 3: Downstream task: link prediction

| Algorithm | Contrast-Reg | Cora | Citeseer | Pubmed | Wiki |
|---|---|---|---|---|---|
| GCN–neg | | $92.40_{\pm0.51}$ | $92.27_{\pm0.90}$ | $97.24_{\pm0.19}$ | $93.27_{\pm0.31}$ |
| node2vec | | $86.33_{\pm0.87}$ | $79.60_{\pm1.58}$ | $81.74_{\pm0.57}$ | $92.41_{\pm0.35}$ |
| DGI | | $93.62_{\pm0.98}$ | $95.03_{\pm1.73}$ | $97.24_{\pm0.13}$ | $95.55_{\pm0.35}$ |
| GMI | | $91.31_{\pm0.88}$ | $92.23_{\pm0.80}$ | $95.14_{\pm0.25}$ | $95.30_{\pm0.29}$ |
| ML-GCN | | $94.81_{\pm0.55}$ | $96.38_{\pm0.63}$ | $92.86_{\pm0.26}$ | $89.84_{\pm0.69}$ |
| GRACE | | $92.82_{\pm0.65}$ | $93.14_{\pm0.73}$ | $96.05_{\pm0.91}$ | $87.72_{\pm1.16}$ |
| LC | ✓ | $94.61_{\pm0.64}$ | $95.63_{\pm0.88}$ | $\mathbf{97.26}_{\pm0.15}$ | $\mathbf{96.28}_{\pm0.21}$ |
| ML-GCN | ✓ | $\mathbf{95.18}_{\pm0.31}$ | $\mathbf{96.83}_{\pm0.55}$ | $95.10_{\pm0.22}$ | $92.70_{\pm0.56}$ |
| GRACE | ✓ | $93.60_{\pm0.57}$ | $93.24_{\pm0.62}$ | $96.60_{\pm1.56}$ | $90.82_{\pm0.99}$ |

Table 4: Pretraining

| Algorithm | Contrast-Reg | Reddit | ogbn-products |
|---|---|---|---|
| No pretraining | | $90.44_{\pm1.62}$ | $84.69_{\pm0.79}$ |
| DGI | | $92.09_{\pm1.05}$ | $86.37_{\pm0.19}$ |
| GMI | | $92.13_{\pm1.16}$ | $86.14_{\pm0.16}$ |
| ML | ✓ | $92.18_{\pm0.97}$ | $86.28_{\pm0.20}$ |
| LC | ✓ | $\mathbf{92.52}_{\pm0.55}$ | $\mathbf{86.45}_{\pm0.13}$ |

### 6.2.2 Graph Clustering

We assess clustering performance using three metrics: accuracy (Acc), normalized mutual information (NMI), and F1-macro (F1), following the previous literature (Xia et al., 2014). Higher values indicate better clustering performance. We compare our methods with DGI, node2vec, GMI, and AGC (Zhang et al., 2019) on the Cora, Citeseer, and Wiki datasets. AGC is a state-of-the-art graph clustering algorithm that leverages high-order graph convolution for attribute graph clustering. For all models and datasets, we employ $k$-means to cluster both the labels and representations of nodes. The clustering results of labels are considered as the ground truth. To reduce dimensionality, we apply PCA to the representations before using $k$-means, since high dimensionality can negatively impact clustering (Chen, 2018). The random seed setting for model training is consistent with that in the node classification task. To minimize randomness, we set the clustering random seed from 0 to 4 and compute the average result for each learned representation. Table 2 presents improved results with and without PCA for each cell. Our algorithms, particularly ours (ML), exhibit superior performance in all cases, demonstrating the effectiveness of Contrast-Reg. It is worth noting that the superior results of ours (ML) compared to ours (LC) suggest that attributes play a crucial role in clustering, as graph clustering is applied to attribute graphs.

### 6.2.3 Link Prediction

In order to circumvent the data linkage issue in link prediction, we employ an inductive setting for graph representation learning. We randomly extract induced subgraphs (comprising 85% of the edges) from each original graph for training both the representation learning model and the link predictor, while reserving the remaining edges for validation and testing (10% for the test edge set and 5% for the validation edge set). We assess performance across 25 (5x5) different runs, utilizing a fixed-seed random split scheme with five distinct induced subgraphs and five fixed-seed runs (ranging from 0 to 4). We compare our results with the baseline algorithms, including DGI, GMI, node2vec, ML-GCN (Shiao et al., 2022), GRACE (Zhu et al., 2020), and unsupervised GCN (GCN-neg in Table 3). In addition, we incorporate Contrast-Reg into LC, ML-GCN, and GRACE against the aforementioned baselines. Our results, presented in Table 3, demonstrate that Contrast-Reg consistently improves the performance of graph contrastive learning models, as evidenced by comparing the results of ML-GCN and GRACE with and without Contrast-Reg. Furthermore, we achieve the state-of-the-art results with the algorithms incorporating Contrast-Reg.

Table 5: Graph contrastive learning with regularization

| Algorithm | Regularization | Cora | Wiki | Computers | Reddit |
|---|---|---|---|---|---|
| ML | | $73.22 \pm 0.77$ | $58.70 \pm 1.51$ | $77.08 \pm 2.48$ | $94.33 \pm 0.07$ |
| ML | $\ell_2$-normalization | $80.14 \pm 0.92$ | $61.83 \pm 1.36$ | $80.80 \pm 1.45$ | $94.07 \pm 0.19$ |
| ML | Weight-decay | $81.65 \pm 0.42$ | $63.67 \pm 1.47$ | $79.09 \pm 0.32$ | $94.34 \pm 0.06$ |
| ML | Contrast-Reg | $\mathbf{82.65} \pm 0.57$ | $\mathbf{67.20} \pm 0.96$ | $\mathbf{82.11} \pm 1.47$ | $\mathbf{94.38} \pm 0.04$ |
| LC | | $79.73 \pm 0.75$ | $65.14 \pm 1.48$ | $79.80 \pm 1.49$ | $94.42 \pm 0.03$ |
| LC | $\ell_2$-normalization | $81.09 \pm 0.59$ | $63.96 \pm 0.64$ | $80.73 \pm 1.36$ | $94.20 \pm 0.06$ |
| LC | Weight-decay | $81.94 \pm 0.44$ | $65.17 \pm 1.34$ | $81.89 \pm 1.58$ | $\mathbf{94.44} \pm 0.03$ |
| LC | Contrast-Reg | $\mathbf{82.33} \pm 0.41$ | $\mathbf{69.19} \pm 1.13$ | $\mathbf{81.98} \pm 1.52$ | $94.43 \pm 0.03$ |

### 6.2.4 Pretraining

We further assess the performance of Contrast-Reg in the context of the pretrain-finetune paradigm. For the Reddit dataset, we naturally partition the data by time, pretraining the models using the first 20 days. We generate an induced subgraph based on the pretraining nodes and divide the remaining data into three parts: the first part produce a new subgraph for fine-tuning the pre-trained model and training the classifier, while the second and third parts are designated for validation and testing. For the ogbn-products dataset, we split the data according to node ID, pretraining the models using a subgraph generated by the initial 70% of the nodes. The data splitting scheme for the remaining data mirrors that of the Reddit dataset. We conduct baseline experiments on DGI and GMI, employing the same GraphSAGE with GCN-aggregation encoder as in our model. Table 4 reveals that pretraining the model facilitates convergence to a more robust representation model with reduced variance, and Contrast-Reg can enhance the transferability of the pre-trained model.

### 6.3 Discussion of Contrast-Reg

To evaluate the benefits of Contrast-Reg, we conduct experiments comparing it to two approaches, specifically, $\ell_2$-normalization and weight decay, on four networks from different domains, with a focus on node classification task performance. It is important to note that some regularization techniques, such as those mentioned in Section 2, are not explicitly designed to address the miscalibration problem. $\ell_2$-normalization (Chen et al., 2020) mitigates the potential risk of the expectation of the prediction value for randomly sampled pairs $\mathbb{E}_{(v,v')}[\sigma(v \cdot v')]$ exploding by explicitly eliminating the embedding norm for each node's embeddings to tackle the miscalibration problem, while weight decay strives to achieve the same result by implicitly restricting the gradient descent step length. Table 5 presents a comparison of the accuracy achieved by different contrastive learning algorithms, **ML** and **LC**, when incorporating $\ell_2$-normalization, weight decay, and Contrast-Reg, as opposed to using the vanilla algorithms. The results indicate that integrating $\ell_2$-normalization, weight decay, and Contrast-Reg into graph contrastive learning algorithms improves the accuracy of downstream tasks, suggesting that addressing miscalibration enhances the generalization of learned embeddings for downstream tasks. However, the performance gains provided by Contrast-Reg exceed those of $\ell_2$-normalization and weight decay. This implies that while alternative algorithms exist to address miscalibration, Contrast-Reg emerges as the most effective method for improving the generalization of learned embeddings for downstream tasks. It is crucial to acknowledge the existence of other algorithms that may also contribute to mitigating the miscalibration problem, and further research is needed to explore and compare their effectiveness.

### 6.4 Generality of Contrast-Reg

The ablation study of Contrast-Reg on the GAT and GIN backbone is presented in Table 6. The results demonstrate that Contrast-Reg consistently enhances graph contrastive learning performance across various backbones and graph contrastive learning algorithms. This conclusion is further supported by the comparison of various models, including GCA with GCA+Contrast-Reg in Table 1, GRACE with GRACE+Contrast-Reg, as well as ML-GCN with ML-GCN+Contrast-Reg in Table 3. These experiments support that Contrast-

Table 6: GAT/GIN as an encode backbone w/ and w/o Contrast-Reg

| Algorithm | Backbone | Contrast-Reg | Cora | Citeseer | Pubmed | Wiki | Computers |
|---|---|---|---|---|---|---|---|
| ML | GAT | | $76.90_{\pm1.20}$ | $70.43_{\pm0.31}$ | $75.05_{\pm0.51}$ | $69.02_{\pm1.89}$ | $78.47_{\pm2.60}$ |
| ML | GAT | ✓ | $\mathbf{81.56}_{\pm0.94}$ | $\mathbf{72.73}_{\pm0.44}$ | $\mathbf{78.50}_{\pm0.63}$ | $\mathbf{69.41}_{\pm0.85}$ | $\mathbf{80.41}_{\pm1.96}$ |
| LC | GAT | | $74.30_{\pm1.10}$ | $69.30_{\pm0.42}$ | $72.50_{\pm0.49}$ | $61.55_{\pm1.89}$ | $73.27_{\pm5.98}$ |
| LC | GAT | ✓ | $\mathbf{80.87}_{\pm0.96}$ | $\mathbf{72.69}_{\pm0.52}$ | $\mathbf{78.50}_{\pm0.63}$ | $\mathbf{69.92}_{\pm0.80}$ | $\mathbf{80.28}_{\pm2.09}$ |
| ML | GIN | | $77.78_{\pm0.98}$ | $70.78_{\pm0.68}$ | $80.12_{\pm0.50}$ | $63.50_{\pm0.93}$ | $70.69_{\pm1.82}$ |
| ML | GIN | ✓ | $\mathbf{81.04}_{\pm0.94}$ | $\mathbf{71.17}_{\pm0.45}$ | $\mathbf{80.23}_{\pm0.69}$ | $\mathbf{66.77}_{\pm0.53}$ | $\mathbf{77.85}_{\pm1.00}$ |
| LC | GIN | | $80.23_{\pm0.61}$ | $71.46_{\pm0.44}$ | $78.46_{\pm0.51}$ | $61.95_{\pm0.81}$ | $70.61_{\pm1.96}$ |
| LC | GIN | ✓ | $\mathbf{81.38}_{\pm0.58}$ | $\mathbf{71.59}_{\pm0.36}$ | $\mathbf{78.51}_{\pm0.46}$ | $\mathbf{67.14}_{\pm0.44}$ | $\mathbf{76.79}_{\pm1.91}$ |

Reg serves as an effective regularization term for graph contrastive learning algorithms that utilize different similarity definitions and GNN encoder backbones. Furthermore, we have also conducted experiments on a graph autoencoder algorithm, GraphMAE (Hou et al., 2022), which is also graph self-supervised learning. The empirical results presented in Table 1 reveal that when combined with Contrast-Reg, GraphMAE achieves superior empirical performance, thus highlighting the potential benefits of Contrast-Reg for graph autoencoder methods. A comprehensive analysis of the impact of Contrast-Reg on graph autoencoder algorithms is an interesting direction and will be left as future work, which includes the theoretical investigations and empirical evaluations involving a broader range of graph autoencoder algorithms (Hou et al., 2022; 2023; Tan et al., 2023; Li et al., 2023).

## 7 Conclusions

We adapted expected calibration error (ECE) to the graph contrastive learning framework and identified the key factors that influence the discrepancy between the prediction in unsupervised training and the accuracy in downstream tasks. Motivated by ECE, we proposed a novel regularization regularizer, Contrast-Reg, to ensure that decreasing the contrastive loss leads to better performance in the downstream tasks. Our theoretical and empirical results both demonstrate the effectiveness of Contrast-Reg in improving the generalizability of the base graph contrastive learning model and achieving superior performance across different existing graph contrastive learning algorithms.

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

Table 7: Impact of Contrast-Reg on ECE Value

|  | Contrast-Reg | Cora | Citeseer | Pubmed |
|---|---|---|---|---|
| ML |  | 0.477 | 0.540 | 0.399 |
| ML | ✓ | **0.413** | **0.525** | **0.273** |
| LC |  | 0.477 | 0.537 | 0.416 |
| LC | ✓ | **0.437** | **0.524** | **0.274** |

## A  Details of the analysis

### A.1  A through analysis of Claim 4.1

Considering that positive sampling is based on the calculated distance between pairwise embeddings, we can express the following relationship:

$$\mathbb{E}_{acc(v,v'_+)=1}[p(v,v'_+)] = \mathbb{E}_{acc(v,v'_+)=0}[p(v,v'_+)] = \mathbb{E}_{\text{topk } \sigma(h_v \cdot h_{v'})}[\sigma(h_v \cdot h_{v'})] \tag{9}$$

Here, $\mathbb{E}_{\text{topk } \sigma(h_v \cdot h_{v'})}[\sigma(h_v \cdot h_{v'})]$ represents the expectation of the top $k$ pairs $(v, v')$. When the loss converges, $\mathbb{E}_{\text{topk } \sigma(h_v \cdot h_{v'})}[\sigma(h_v \cdot h_{v'})] \to 1$.

Furthermore, by considering the definition in Equation 4 and the fact that negative samples are uniformly sampled, we can write:

$$\mathbb{E}_{acc(v,v'_-)=0}[p(v,v'_-)] = \mathbb{E}_{acc(v,v'_-)=1}[p(v,v'_-)] = 1 - \mathbb{E}_{(v,v')}[\sigma(v,v')]. \tag{10}$$

Thus, Equation 6 can be reformulated as:

$$\begin{aligned}
\text{ECE} = & r^+(1 - 2\mathbb{E}_{\text{topk } \sigma(h_v \cdot h_{v'})}[\sigma(h_v \cdot h_{v'})])q^+ + r^-(1 - 2q^-)\mathbb{E}[\sigma(h_v \cdot h_{v'})] + r^- q^- \\
& + r^+\mathbb{E}_{\text{topk } \sigma(h_v \cdot h_{v'})}[\sigma(h_v \cdot h'_v)].
\end{aligned} \tag{11}$$

When the loss converges, $\mathbb{E}_{\text{topk } \sigma(h_v \cdot h_{v'})}[\sigma(h_v \cdot h'_v)] \to 1$, indicating that the ECE value is negatively correlated with the probability $q^+$ of $v'_+$. In addition, negative samples are uniformly sampled, so that $q^- = 1/K$, where $K$ represents the number of classes when $K > 2$. As a result, ECE is positively correlated with the expectation of the confidence value for randomly sampled pairs $\mathbb{E}_{(v,v')}[\sigma(h_v, h_{v'})]$.

### A.2  Contrast-Reg Leads to a Decreased ECE Value

We empirically investigate the impact of loss convergence on the ECE value, with and without Contrast-Reg, across various datasets and contrastive strategies, in Table 7. The results demonstrate that Contrast-Reg leads to a decrease in ECE values for all tested datasets and contrastive losses. This reduction in ECE indicates that Contrast-Reg promotes better alignment with the performance of downstream tasks while minimizing the training loss, ensuring that minimizing the contrastive loss with Contrast-Reg results in high-quality representations.

### A.3  Explanation of Contrast-Reg's Impact on the Term $s(f)$

In this section, we provide an explanation for why Contrast-Reg leads to a decrease in the upper bound of $s(f)$. Firstly, we will prove that minimizing Eq. (7) results in a decrease in $\text{Var}(\|h_i\|)$ when $h_i^T W \mathbf{r} > c$, as stated in Theorem 2. Then, based on the assumption that models with lower $\text{Var}(\|h_i\|)$ inherently favor lower values of $\mathbb{E}[\|h_i\|]$, both $\mathbb{E}_{v_i \sim D_c}[\|h_i\|^2]$ and $\sigma_{max}(M(f, c))$ will decrease. Consequently, the upper bound of $s(f)$ decreases with the implementation of Contrast-Reg. In the subsequent proof, $h_i = f(x)$, and the notations $h_i$ and $f(x)$ may be used interchangeably for the sake of presentation clarity.

**Theorem 2.** *Minimizing Eq. (7) induces the decrease in* $\text{Var}(\|h_i\|)$ *when* $h_i^T W \mathbf{r} > c$.

*Proof.* We minimize $\mathcal{L}_{reg}$ by gradient descent with learning rate $\beta$.

$$\frac{\partial}{\partial f(x)} \mathcal{L}_{reg} = -\sigma(-f(x)^T W \mathbf{r}) W \mathbf{r} \tag{12}$$

The embedding of $f(x)$ is updated as the following after adding $\mathcal{L}_reg$:

$$f(x) \leftarrow f(x) + \beta \left( \sigma(-f(x)^T W\mathbf{r})W\mathbf{r} \right) \tag{13}$$

Eq. (13) shows that in every optimization step, $f(x)$ extends by $\beta\sigma(-f(x)^T W\mathbf{r}) \|W\mathbf{r}\|$ along $\mathbf{r}_0 := \frac{W\mathbf{r}}{\|W\mathbf{r}\|}$. If we do orthogonal decomposition for $f(x)$ along $\mathbf{r}_0$ and its unit orthogonal hyperplain $\Pi(\mathbf{r}_0)$, $f(x) = \left(f(x)^T\mathbf{r}_0\right)\mathbf{r}_0 + \left(f(x)^T\Pi(\mathbf{r}_0)\right)\Pi(\mathbf{r}_0)$. Thus we have

$$\|f(x)\| = \sqrt{(f(x)^T\mathbf{r}_0)^2 + (f(x)^T\Pi(\mathbf{r}_0))^2}. \tag{14}$$

The projection of $f(x)$ along $\mathbf{r}_0$ is $f(x)^T\mathbf{r}_0 = \frac{f(x)^T W\mathbf{r}}{\|W\mathbf{r}\|}$, while the projection of $f(x)$ plus the Contrast-Reg update along $\mathbf{r}_0$ is

$$\left(f(x)^T\mathbf{r}_0\right)_{reg} = \frac{f(x)^T W\mathbf{r}}{\|W\mathbf{r}\|} + \frac{\beta}{1 + e^{f(x)^T W\mathbf{r}}}\|W\mathbf{r}\|.$$

Note that $\left(f(x)^T\Pi(\mathbf{r}_0)\right)_{reg} = f(x)^T\Pi(\mathbf{r}_0)$.

Based on Lemma 1 and Eq. (14),

when $\beta\|W\mathbf{r}\|^2 \leq 1$ and $f(x)^T W\mathbf{r} > 1.5$, we have

$$\mathrm{Var}\left(\left\|(f(x))_{reg}\right\|\right) < \mathrm{Var}\left(\|f(x)\|\right). \tag{15}$$

$\square$

**Lemma 1.** *For a random variable $X \in [1.5, +\infty)$, a constant $\tau \in (0, 1]$ and a constant $c^2$, we have*

$$\mathrm{Var}\left(\sqrt{(X + \frac{\tau}{1 + e^X})^2 + c^2}\right) < \mathrm{Var}\left(\sqrt{X^2 + c^2}\right). \tag{16}$$

*Proof.* First, we consider

$$h(x) = \sqrt{(x + \frac{\tau}{1 + e^x})^2 + c^2} - \sqrt{x^2 + c^2},$$

where $h(x)$ is strictly decreasing in $[x_0, +\infty)$ and strictly increasing in $(-\infty, x_0]$, and $x_0$ is the solution of $h'(x) = \frac{\mathrm{d}h(x)}{\mathrm{d}x} = 0$. Thus, we can approximate the range of $x_0 \in (0, 1.5)$ by the fact that $h'(0)h'(1.5) < 0$ for all $\tau$ and $c^2$.

Thus, for $x > y \geq 1.5$,

$$\sqrt{(x + \frac{\tau}{1 + e^x})^2 + c^2} - \sqrt{x^2 + c^2} < \sqrt{(y + \frac{\tau}{1 + e^y})^2 + c^2} - \sqrt{y^2 + c^2}$$

and since $(x + \frac{\tau}{1+e^x})$ is monotonically increasing, we get

$$0 < \sqrt{(x + \frac{\tau}{1 + e^x})^2 + c^2} - \sqrt{(y + \frac{\tau}{1 + e^y})^2 + c^2} < \sqrt{x^2 + c^2} - \sqrt{y^2 + c^2}.$$

When $y > x \geq 1.5$,

$$\sqrt{x^2 + c^2} - \sqrt{y^2 + c^2} < \sqrt{(x + \frac{\tau}{1 + e^x})^2 + c^2} - \sqrt{(y + \frac{\tau}{1 + e^y})^2 + c^2} < 0.$$

Further, we assume that $X$ and $Y$ are i.i.d. random variables sampled from $[1.5, +\infty)$,

$$
\begin{aligned}
&\mathrm{Var}\left(\sqrt{(X + \frac{\tau}{1+e^X})^2 + c^2}\right)\\
=&\frac{1}{2} \times \mathbb{E}_{X,Y}\left[\left(\sqrt{(X + \frac{\tau}{1+e^X})^2 + c^2} - \sqrt{(Y + \frac{\tau}{1+e^Y})^2 + c^2}\right)^2\right]\\
=&\frac{1}{2} \times \int \left(\sqrt{(x + \frac{\tau}{1+e^x})^2 + c^2} - \sqrt{(y + \frac{\tau}{1+e^y})^2 + c^2}\right)^2 p(x)p(y)\mathrm{d}x\mathrm{d}y\\
<&\frac{1}{2} \times \int \left(\sqrt{x^2 + c^2} - \sqrt{y^2 + c^2}\right)^2 p(x)p(y)\mathrm{d}x\mathrm{d}y\\
=&\mathrm{Var}(\sqrt{X^2 + c^2})
\end{aligned}
$$

$\square$

**Remark 1.** $\beta \|W\mathbf{r}\|^2 \leq 1$, *which is the condition of Eq. (15) , is not difficult to satisfy, since the magnitude of $\mathbf{r}$ could be tuned. In practice, $\mathbf{r} \in (0, 1]$ can fit in all our experiments.*

**Remark 2.** *The range of $f(x)^T w\mathbf{r}$ in Theorem 2 is not a tight bound for $x_0$ in Lemma 1. Since when Eq. (7) converges, $f(x)^T W\mathbf{r}$ is much larger than 1.5 for almost all the samples empirically, we prove the case for $f(x)^T w\mathbf{r} \in [1.5, +\infty)$.*

### A.4 Comprehensive Explanation of Theorem 1: Notations and Proof

To formally analyze the behavior of contrastive learning, we introduce the following concepts as the previous literature (Saunshi et al., 2019) does.

- *Latent classes*: Data are considered as drawn from latent classes $\mathcal{C}$ with distribution $\rho$. Further, distribution $\mathcal{D}_c$ is defined over feature space $\mathcal{X}$ that is associated with a class $c \in \mathcal{C}$ to measure the relevance between $x$ and $c$.

- *Semantic similarity*: Positive samples are drawn from the same latent classes, with distribution

$$
\mathcal{D}_{sim}(x, x^+) = \mathbb{E}_{c \in \rho}\left[\mathcal{D}_c(x)\mathcal{D}_c(x^+)\right], \tag{17}
$$

  while negative samples are drawn randomly from all possible data points, i.e., the marginal of $\mathcal{D}_{sim}$, as

$$
\mathcal{D}_{neg}(x^-) = \mathbb{E}_{c \in \rho}\left[\mathcal{D}_c(x^-)\right] \tag{18}
$$

- *Supervised tasks*: Denote $K$ as the number of negative samples. The object of the supervised task, i.e., feature-label pair $(x, c)$, is sampled from

$$
\mathcal{D}_{\mathcal{T}}(x, c) = \mathcal{D}_c(x)\mathcal{D}_{\mathcal{T}}(c),
$$

  where $\mathcal{D}_{\mathcal{T}}(c) = \rho(c | c \in \mathcal{T})$, and $\mathcal{T} \subseteq \mathcal{C}$ with $|\mathcal{T}| = K + 1$.
  Mean classifier $W^\mu$ is naturally imposed to bridge the gap between the representation learning performance and linear separability of learn representations, as

$$
W_c^\mu := \mu_c = \mathbb{E}_{x \sim \mathcal{D}_c}[f(x)].
$$

- *Empirical Rademacher complexity*: Suppose $\mathcal{F} : \mathcal{X} \to [1, 0]$. Given a sample $\mathcal{S}$,

$$
\mathcal{R}_{\mathcal{S}}(\mathcal{F}) = \mathbb{E}_{\vec{e}}\left[\sup_{f \in \mathcal{F}} \vec{e}^T f(\mathcal{S})\right],
$$

  where $\vec{e} = (e_1, \cdots, e_m)^T$, with $e_i$ are independent random variables taking values uniformly from $\{-1, +1\}$.

In addition, the theoretical framework (Saunshi et al., 2019) makes an assumption: encoder $f$ is bounded, i.e., $\max_{x \in \mathcal{X}} \|f(x)\| \leq R^2$, $R \in \mathbb{R}$.

To prove Theorem 1, we first list some key lemmas.

**Lemma 2.** *For all $f \in \mathcal{F}$,*

$$\mathcal{L}_{sup}^{\mu}(f) \leq \frac{1}{1-\tau}(\mathcal{L}_{nce}(f) - 2\tau \log 2). \tag{19}$$

This bound connects contrastive representation learning algorithms and its supervised counterpart. This lemma is achieved by Jensen's inequality. The details are given in Appendix A.6.

**Lemma 3.** *With probability at least $1 - \delta$ over the set $\mathcal{S}$, for all $f \in \mathcal{F}$,*

$$\mathcal{L}_{nce}(\hat{f}) \leq \mathcal{L}_{nce}(f) + Gen_M. \tag{20}$$

This bound guarantees that the chosen $\hat{f} = \arg\min_{f \in \mathcal{F}} \mathcal{L}_{nce}^{\mu}$ cannot be too much worse than $f^* = \arg\min_{f \in \mathcal{F}} \mathcal{L}_{nce}$. The proof applies Rademacher complexity of the function class (Mohri et al., 2018) and vector-contraction inequality (Maurer, 2016). More details are given in Appendix A.7.

**Lemma 4.** $\mathcal{L}_{nce}^{=}(f) \leq 4s(f) + 2\log 2$.

This bound is derived by the loss caused by both positive and negative pairs that come from the same class, i.e., class collision. The proof uses Bernoulli's inequality (details in Appendix A.8).

*Proof to Theorem 1.* Combining Lemma 2 and Lemma 3, we obtain with probability at least $1 - \delta$ over the set $\mathcal{S}$, for all $f \in \mathcal{F}$,

$$\mathcal{L}_{sup}^{\mu}(\hat{f}) \leq \frac{1}{1-\tau}\left(\mathcal{L}_{nce}(f) + Gen_M\right) \tag{21}$$

Then, we decompose $\mathcal{L}_{nce} = \tau \mathcal{L}_{nce}^{=}(f) + (1-\tau)\mathcal{L}_{nce}^{\neq}(f)$, apply Lemma 4 to Eq. (21), and obtain the result of Theorem 1 $\qquad\square$

## A.5 Contrastive Learning with NCEloss

The contrastive loss is

$$\mathcal{L}_{un} \coloneqq \mathop{\mathbb{E}}_{\substack{(x,x^+) \sim \mathcal{D}_{sim}, \\ (x_1^-, \cdots, x_K^-) \sim \mathcal{D}_{neg}}} \left[\ell(\{f(x)^T(f(x^+) - f(x_i^-))\}_{i=1}^K)\right],$$

where $\ell$ can be the hinge loss as $\ell(\mathbf{v}) = \max\{0, 1 + \max_i\{-\mathbf{v}_i\}\}$ or the logistic loss as $\ell(\mathbf{v}) = \log_2(1 + \sum_i \exp(-\mathbf{v}_i))$. And its supervised counterpart is defined as

$$\mathcal{L}_{sup}^{\mu} \coloneqq \mathop{\mathbb{E}}_{(x,c) \sim \mathcal{D}_{\mathcal{T}(x,c)}} \left[\ell\left(\left\{f(x)^T \mu_c - f(x)^T \mu_{c'}\right\}_{c' \neq c}\right)\right].$$

A more powerful loss function, NCEloss, used in the previous literature (Velickovic et al., 2019; Yang et al., 2020; Mnih & Teh, 2012; Dyer, 2014), can be framed as

$$\mathcal{L}_{nce} \coloneqq$$
$$-\mathop{\mathbb{E}}_{\substack{(x,x^+) \sim \mathcal{D}_{sim}, \\ (x_1^-, \cdots, x_K^-) \sim \mathcal{D}_{neg}}} \left[\log \sigma(f(x)^T f(x^+)) + \sum_{k=1}^K \log \sigma(-f(x)^T f(x_k^-))\right], \tag{22}$$

and its empirical counterpart with $M$ samples $\left(x_i, x_i^+, x_{i1}^-, \cdots, x_{iK}^-\right)_{i=1}^M$ is given as

$$\hat{\mathcal{L}}_{nce} \coloneqq -\frac{1}{M} \sum_{i=1}^M \left[\log \sigma(f(x_i)^T f(x_i^+)) + \sum_{k=1}^K \log \sigma(-f(x_i)^T f(x_{ij}^-))\right], \tag{23}$$

where $\sigma(\cdot)$ is the sigmoid function.

For its supervised counterpart, it is exactly the cross entropy loss for the $(K+1)$-way multi-class classification task:

$$\mathcal{L}_{sup}^{\mu} \coloneqq -\mathop{\mathbb{E}}_{(x,c) \sim \mathcal{D}_{\mathcal{T}(x,c)}} \left[\log \sigma(f(x)^T \mu_c) + \log \sigma(-f(x)^T \mu_{c'}) \,|\, c' \neq c\right]. \tag{24}$$

## A.6 Proof of Lemma 2

First, we prove that $\ell(f(x^+), \{f(x_i^-)\}) = -(\log \sigma(f(x)^T f(x^+)) + \sum_{i=1}^K \log(\sigma(f(x)^T f(x^-)))$ is convex w.r.t. $f(x^+), f(x_i^-), \cdots, f(x_K^-)$. Consider that $\ell_1(z) = -\log \sigma(z)$ and $\ell_2(z) = -\log \sigma(-z)$ are both convex functions since $\ell_1'' > 0$ and $\ell_2'' > 0$ for $z \in \mathbb{R}$. Given $f(x) \in \mathbb{R}$, $z^+ = f(x)^T f(x^+)$ and $z^- = f(x)^T f(x^+)$ are affine transformation w.r.t. $f(x^+)$ and $f(x^-)$. Thus, when $f(x)$ is fixed, $\ell_1(f(x^+)) = -\log \sigma(f(x)^T f(x^+))$ and $\ell_2(f(x^-)) = -\log \sigma(-f(x)^T f(x^-))$ are convex functions. As $\ell_1 > 0$ and $\ell_2 > 0$, we obtain $\ell(f(x^+), \{f(x_i^-)\}) = -(\log \sigma(f(x)^T f(x^+)) + \sum_{i=1}^K \log(\sigma(f(x)^T f(x^-))))$ is convex since non-negative weighted sums preserve convexity (Boyd & Vandenberghe, 2014). By the definition of convexity,

$$
\begin{aligned}
\mathcal{L}_{nce}(f) &= \mathbb{E}_{\substack{c^+,c^- \sim \rho^2; \\ x \in \mathcal{D}_{c^+}}} \mathbb{E}_{\substack{x^+ \sim \mathcal{D}_{c^+}; \\ x^- \sim \mathcal{D}_{c^-}}} \left[ \ell(f(x^+), \{f(x_i^-)\}) \right] \\
&\geq \mathbb{E}_{c^+,c^- \sim \rho^2} \mathbb{E}_{x \sim \mathcal{D}_{c^+}} \left[ \ell(f(x)^T \{\mu_{c^+}, \mu_{c^-}\}) \right] \\
&= (1-\tau) \mathcal{L}_{sup}^\mu(f) + \tau \mathbb{E}_{c^+ \sim \rho} \mathbb{E}_{x \sim \mathcal{D}_{c^+}} \left[ -\log \sigma(f(x)^T \mu_{c^+}) - \log \sigma(-f(x)^T \mu_{c^+}) \right] \\
&\geq (1-\tau) \mathcal{L}_{sup}^\mu(f) + 2\tau \log 2
\end{aligned}
$$

## A.7 Generalization bound

Denote

$$
\begin{aligned}
\tilde{\mathcal{F}} = \Big\{ \tilde{f}\left(x_i, x_i^+, x_{i1}^-, \cdots, x_{iK}^-\right) = \\
\left(f(x_i), f(x_i^+), f(x_{i1}^-), \cdots, f(x_{iK}^-)\right) \big| f \in \mathcal{F} \Big\}.
\end{aligned}
$$

Let $q_{\tilde{f}} = h \circ \tilde{f}$, and its function class,

$$
\mathcal{Q} = \left\{ q = h \circ \tilde{f} \big| \tilde{f} \in \tilde{\mathcal{F}} \right\}.
$$

Denote $z_i = \left(x_i, x_i^+, x_{i1}^-, \cdots, x_{iK}^-\right)$, suppose $\ell$ is bounded by $B$, then we can decompose $h = \frac{1}{B} \ell \circ \phi$. Then we have $q_{\tilde{f}}(z_i) = \frac{1}{B} \ell(\phi(\tilde{f}(z_i)))$, where

$$
\begin{aligned}
\phi(\tilde{f}(z_i)) &= \Bigg( \sum_{t=1}^d f(x_i)_t f(x_{i0}^+)_t, \sum_{t=1}^d f(x_i)_t f(x_{i1}^-)_t, \cdots, \\
&\qquad\qquad \sum_{t=1}^d f(x_i)_t f(x_{iK}^-)_t \Bigg) \\
\ell(\mathbf{x}) &= -\left( \log \sigma(x_0) + \sum_{i=1}^K \log \sigma(-x_i) \right).
\end{aligned}
\tag{25}
$$

From Eq. (25), we know that $\phi : \mathbb{R}^{(K+2)d} \to \mathbb{R}^{K+1}$.
Then we will prove that $h$ is $L$-Lipschitz by proving that $\phi$ and $\ell$ are both Lipschitz continuity. First,

$$
\begin{aligned}
\frac{\partial \phi(\tilde{f}(z_i))}{\partial f(x_i)_t} &= f(x_{ik})_t = \begin{cases} f(x_{i0}^+)_t, & k = 0 \\ f(x_{ik}^-)_t, & k = 1, \cdots, K \end{cases} \\
\frac{\partial \phi(\tilde{f}(z_i))}{f(x_{i0}^+)_t} &= f(x_i)_t, \qquad \frac{\partial \phi(\tilde{f}(z_i))}{f(x_{ik}^-)_t} = f(x_i)_t.
\end{aligned}
$$

If we assume $\sum_{t=1}^d f(x_{ik})_t^2 \leq R^2$ and $\sum_{t=1}^d f(x_i)_t^2 \leq R^2$,

$$
\begin{aligned}
\|J\|_F &= \sqrt{\sum_{t=1}^d f(x_{i0}^+)_t^2 + \sum_{k=1}^K \sum_{t=1}^d f(x_{ik}^-)_t^2 + (K+1) \sum_{t=1}^d f(x_i)_t^2} \\
&\leq \sqrt{2(K+1)R^2} = \sqrt{2(K+1)}R
\end{aligned}
$$

---

**Algorithm 2:** ML

---

**Parameter:** Parameters of an (additional) GNN layer $g$.

**1 Function** Contrast($\mathcal{C}$, $\mathcal{G}$, $\mathcal{X}$, $f$):

**2** $\quad$ Let $g(x_i)$ be the representation of $x_i$ by stacking $g$ upon $f$;

**3** $\quad$ Randomly pick a negative node $x_i^-$ from $\mathcal{V}$ for each $x_i \in \mathcal{C}$;

**4** $\quad$ **return** $\left\{ (g(x_i), f(x_i), f(x_i^-)) \right\}_{x_i \in \mathcal{G}}$;

**5 end**

**6 Function** SeedSelect($\mathcal{G}$, $\mathcal{X}$, $f$, *epoch*):

**7** $\quad$ **return** $\mathcal{V}$;

**8 end**

---

Combining $\|J\|_2 \le \|J\|_F$, we obtain that $\phi$ is $\sqrt{2(K+1)}R$-Lipschitz. Similarly, $\ell$ is $\sqrt{K+1}$-Lipschitz. Since we assume that the inner product of embedding is no more than $R^2$. Thus, $l$ is bounded by $B = -(K+1)\log(\sigma(-R^2))$. Above all, $h$ is $L$-Lipschitz with $L = \frac{\sqrt{2}(K+1)R}{B}$. Applying vector-contraction inequality (Maurer, 2016), we have

$$\mathbb{E}_{\sigma \sim \{\pm 1\}^M}[\sup_{\tilde{f} \in \tilde{\mathcal{F}}} \langle \sigma, (h \circ \tilde{f})_{|\mathcal{S}} \rangle] \le \sqrt{2}L \mathbb{E}_{\sigma \sim \{\pm 1\}^{(K+1)dM}}[\sup_{\tilde{f} \in \tilde{\mathcal{F}}} \langle \sigma, \tilde{f}_{|\mathcal{S}} \rangle].$$

If we write it in Rademacher complexity manner, we have

$$\mathcal{R}_{\mathcal{S}}(\mathcal{Q}) \le \frac{2(K+1)R}{B} \mathcal{R}_{\mathcal{S}}(\mathcal{F}).$$

Applying generalization bounds based on Rademacher complexity (Mohri et al., 2018) to $q \in \mathcal{Q}$. For any $\delta > 0$, with the probability of at least $1 - \frac{\delta}{2}$,

$$\begin{aligned}
\mathbb{E}[q(\mathbf{z})] &\le \frac{1}{M} \sum_{i=1}^{M} q(\mathbf{z}_i) + \frac{2\mathcal{R}_{\mathcal{S}}(\mathcal{Q})}{M} + 3\sqrt{\frac{\log \frac{4}{\delta}}{2M}} \\
&\le \frac{1}{M} \sum_{i=1}^{M} q(\mathbf{z}_i) + \frac{4(K+1)R\mathcal{R}_{\mathcal{S}}(\mathcal{F})}{BM} + 3\sqrt{\frac{\log \frac{4}{\delta}}{2M}}.
\end{aligned}$$

Thus for any $f$,

$$\mathcal{L}_{nce}(f) \le \tilde{\mathcal{L}}_{nce}(f) + \frac{4(K+1)R\mathcal{R}_{\mathcal{S}}(\mathcal{F})}{M} + 3B\sqrt{\frac{\log \frac{4}{\delta}}{2M}}. \tag{26}$$

Let $\hat{f} = \arg\min_{f \in \mathcal{F}} \tilde{\mathcal{L}}_{nce}(f)$ and $f^* = \arg\min_{f \in \mathcal{F}} \mathcal{L}_{nce}(f)$. By Hoeffding's inequality, with probability of $1 - \frac{\delta}{2}$,

$$\tilde{\mathcal{L}}_{nce}(f^*) \le \mathcal{L}_{nce}(f^*) + B\sqrt{\frac{\log \frac{2}{\delta}}{2M}} \tag{27}$$

Substituting $\hat{f}$ into Eq. (26), combining $\tilde{\mathcal{L}}_{nce}(\hat{f}) \le \mathcal{L}_{nce}(f^*)$ and applying union bound, with probability of at most $\delta$

$$\begin{aligned}
\mathcal{L}_{nce}(\hat{f}) &\le \tilde{\mathcal{L}}_{nce}(\hat{f}) + \frac{4(K+1)R\mathcal{R}_{\mathcal{S}}(\mathcal{F})}{M} + 3B\sqrt{\frac{\log \frac{4}{\delta}}{2M}} + B\sqrt{\frac{\log \frac{2}{\delta}}{2M}} \\
&\le \mathcal{L}_{nce}(f^*) + \frac{4(K+1)R\mathcal{R}_{\mathcal{S}}(\mathcal{F})}{M} + 4B\sqrt{\frac{\log \frac{4}{\delta}}{2M}} \\
&\le \mathcal{L}_{nce}(f) + \frac{4(K+1)R\mathcal{R}_{\mathcal{S}}(\mathcal{F})}{M} - 4(K+1)\log(\sigma(-R^2))\sqrt{\frac{\log \frac{4}{\delta}}{2M}}
\end{aligned} \tag{28}$$

fails. Thus, with probability of at least $1 - \delta$, Eq. (28) holds.

### A.8 Class collision loss

Let $p_i = |f(x)^T f(x_i)|$ and $p = \max_{i \in \{0, 1, \cdots, K\}} p_i$. Considering

$$
\begin{aligned}
\mathcal{L}_{nce}^{=}(f) &= -\mathbb{E}\left[\log \sigma(f(x)^T f(x_0^+)) + \sum_{i=1}^{K} \log \sigma(-f(x)^T f(x_i^-)))\right] \\
&= \mathbb{E}\left[\log(1 + e^{-f(x)^T f(x_0^+)}) + \sum_{i=1}^{K} \log(1 + e^{f(x)^T f(x_i^-)})\right] \\
&\leq (K+1)\mathbb{E}\left[\log(1 + e^p)\right] \\
&\leq (K+1)\log 2 + (K+1)\mathbb{E}[p]
\end{aligned}
\tag{29}
$$

Since $x, x_0^+, x_1^-, \cdots, x_K^-$ are sampled i.i.d. from the same class,

$$
\mathbb{E}[p] = \int P[p \geq x]dx = \int (1 - (1 - P[p_0 \geq x])^{K+1}dx.
\tag{30}
$$

Applying Bernoulli's inequality, we have

$$
\begin{aligned}
\mathbb{E}[p] &\leq \int (1 - (1 - (K+1)P[p_0 \geq x]))dx \\
&= \int (K+1)P[p_0 \geq x]dx \\
&= (K+1)\mathbb{E}[p_0] \\
&= (K+1)\mathbb{E}[|f(x)^T f(x_0^+)|] \\
&\leq (K+1)\sqrt{\mathbb{E}[(f(x)^T f(x_0^+))^2]}.
\end{aligned}
\tag{31}
$$

Therefore,

$$
\mathcal{L}_{nce}^{=}(f) \leq (K+1)\log 2 + (K+1)^2 s(f)
\tag{32}
$$

---

**Algorithm 3:** LC

---

**Hyperparameter:** $R$: curriculum update epochs; $k$: the number of candidate positive samples for seed node;

**1 Function** Contrast($\mathcal{C}$, $\mathcal{G}$, $\mathcal{X}$, $f$):

**2**     For $x_i \in \mathcal{C}$, let $\mathcal{N}_i^+$ be the set of $k$ nodes in $\{x_j \in \text{Neighbor}(x_i)\}$ with largest $f(x_i)^T f(x_j)$;

**3**     Randomly pick one positive node $x_i^+$ from $\mathcal{N}_i^+$ for each $x_i \in \mathcal{C}$;

**4**     Randomly pick one negative node $x_i^-$ from $\mathcal{V}$ for each $x_i \in \mathcal{C}$;

**5**     **return** $\left\{(f(x_i), f(x_i^+), f(x_i^-))\right\}_{x_i \in \mathcal{C}}$;

**6 end**

**7 Function** SeedSelect($\mathcal{G}$, $\mathcal{X}$, $f$, *epoch*):

**8**     **if** *epoch % R $\neq$ 1* **then**

**9**       |   **return** the same set of seed nodes $C$ as in the last epoch ;

**10**     **end**

**11**     $p_{i,j} \leftarrow \dfrac{f(i)^T f(j)}{\sum_{k \in \mathcal{V}} f(i)^T f(k)}$ for $i, j \in \mathcal{V}$;

**12**     $H(i) \leftarrow -\sum_{j \in \mathcal{V}}(p_{i,j} \log (p_{i,j}))$ for $i \in \mathcal{V}$;

**13**     **return** $(\lfloor \frac{epoch}{R} \rfloor + 1)\frac{R}{e}|\mathcal{G}|$ nodes with smallest $H$;

**14 end**

---

---

**Algorithm 4:** GCA

---

**Hyperparameter:** two stochastic augmentation functions set $\mathcal{T}$ and $\mathcal{T}'$;

**1 Function** Contrast($\mathcal{C}$, $\mathcal{G}$, $\mathcal{X}$, $f$):

**2**     Sample two stochastic augmentation functions $t \sim \mathcal{T}$ and $t \sim \mathcal{T}'$;

**3**     Generate two graph views $\tilde{G}_1 = t(G)$ and $\tilde{G}_2 = t'(G)$ by performing corruption on $G$;

**4**     **return** $\left\{(f(\tilde{G}_1(x_i)), f(\tilde{G}_2(x_i)), f(\tilde{G}_2(x_j)))\right\}_{x_i, x_i \in \mathcal{G}, x_j \neq x_i}$;

**5 end**

**6 Function** SeedSelect($\mathcal{G}$, $\mathcal{X}$, $f$, *epoch*):

**7**     **return** $\mathcal{V}$;

**8 end**

---

# B    Details of Graph Contrastive Algorithms

Algorithm 1 presents the graph contrastive learning framework and how Contrast-Reg is plugged into it. The functions *SeedSelect* and *Contrast* in the graph contrastive learning framework are responsible for selecting the seed nodes and sampling the positive/negative pairs for these seeds while incorporating various priors for both structural and attribute aspects of the graph. he detailed *SeedSelect* and *Contrast* implementations for ML, LC, GCA are referred to Algorithm 2, 3 and 4.

# C    Supplementary Experiments

## C.1   Plug Contrast-Reg into DiGCL and GDCL

We applied Contrast-Reg to two graph contrastive learning algorithms, namely DiGCL (Tong et al., 2021) and GDCL (Zhao et al., 2021). GDCL employs graph clustering to reduce the number of false-negative samples, whereas DiGCL generates contrastive pairs using Laplacian perturbations to better preserve the structural attributes of directed graphs. In Table 8, we present a comparative analysis of the performance of Contrast-Reg, applied to both algorithms, as well as their performance without Contrast-Reg. We used the official implementations and the parameters of both algorithms, and conducted experiments with random seeds ranging from 0 to 9. Our results demonstrate that Contrast-Reg serves as an effective plugin to these advanced graph contrastive learning algorithms, even when applied to directed graphs (DiGCL).

Table 8: DiGCL & GDCL with and without Contrast-Reg

| Algorithm | Contrast-Reg | Cora-ml | Citeseer | Cora |
|---|---|---|---|---|
| DiGCL | | $76.62_{\pm 1.23}$ | $65.54_{\pm 1.39}$ | |
| DiGCL | ✓ | $\mathbf{79.74}_{\pm 0.62}$ | $\mathbf{68.74}_{\pm 1.13}$ | |
| GDCL | | | $72.98_{\pm 0.73}$ | $85.41_{\pm 0.53}$ |
| GDCL | ✓ | | $\mathbf{76.39}_{\pm 1.16}$ | $\mathbf{85.80}_{\pm 0.46}$ |

Table 9: Datasets

| Dataset | Node # | Edge # | Feature # | Class # |
|---|---|---|---|---|
| Cora (Yang et al., 2016) | 2,708 | 5,429 | 1,433 | 7 |
| Citeseer (Yang et al., 2016) | 3,327 | 4,732 | 3,703 | 6 |
| Pubmed (Yang et al., 2016) | 19,717 | 44,338 | 500 | 3 |
| ogbn-arxiv (Hu et al., 2020a) | 169,343 | 1,166,243 | 128 | 40 |
| Wiki (Yang et al., 2015) | 2,405 | 17,981 | 4,973 | 3 |
| Computers (Shchur et al., 2018) | 13,381 | 245,778 | 767 | 10 |
| Photo (Shchur et al., 2018) | 7,487 | 119,043 | 745 | 8 |
| ogbn-products (Hu et al., 2020a) | 2,449,029 | 61,859,140 | 100 | 47 |
| Reddit (Hamilton et al., 2017) | 232,965 | 114,615,892 | 602 | 41 |

## D  Experiment details

**Dataset statistics**  The datasets we employed encompass citation networks, web graphs, co-purchase networks, and social networks. The detailed dataset statistics are shown in Table 9.

**Hardware Configuration:**  The experiments are conducted on Linux servers installed with an Intel(R) Xeon(R) Silver 4114 CPU @ 2.20GHz, 256GB RAM and 8 NVIDIA 2080Ti GPUs.

**Software Configuration:**  Our models, as well as the DGI, GMI and GCN baselines, were implemented in PyTorch Geometric (Fey & Lenssen, 2019) version 1.4.3, DGL (Wang et al., 2019) version 0.5.1 with CUDA version 10.2, scikit-learn version 0.23.1 and Python 3.6. Our codes and datasets will be made available.

**Hyper-parameters:**  For full batch training, we used 1-layer GCN as the encoder with prelu activation, for mini-batch training, we used a 3-layer GCN with prelu activation. We conducted grid search of different learning rate (from 1e-2, 5e-3, 3e-3, 1e-3, 5e-4, 3e-4, 1e-4) and curriculum settings (including learning rate decay and curriculum rounds) on the fullbatch version. We used 1e-3 or 5e-4 as the learning rate; 10,10,15 or 10,10,25 as the fanouts and 1024 or 512 as the batch size for mini-batch training.

