# OpenReview forum: "Calibrating and Improving Graph Contrastive Learning"
_TMLR — Accepted by TMLR_

### Review · Reviewer_mn1F · 2023-05-01

**Summary Of Contributions:**

In this work, the authors proposed a new and effective regularizer to improve existing graph contrastive learning (GCL) framework.
The proposed regularizer is based on the expected calibration error (ECE) that measures the consistency between model confidence and model performance.
By imposing this regularizer, the authors empirically show that the mis-calibration/bias of GCL caused by the randomly sampled negative pairs can be suppressed.
In theory, the authors demonstrate that the proposed regularizer helps to reduce the expectation of the embeddings’ inner product and thus leads to a tighter generalization error bound of GCL.
In the aspect of methodology, the authors provide two implementations of the regularized GCL and verify their effectiveness in multiple downstream graph learning tasks.



**Audience:**

Yes

**Claims And Evidence:**

Yes

**Requested Changes:**

See the drawbacks and minors shown above

**Strengths And Weaknesses:**

Strengths:

1. The motivation of the proposed method is clearly explained. The mis-calibration is the key challenge of GCL indeed. Therefore, the proposed work makes a significant contribution to the community of GCL.

2. The idea of designing an ECE-based regularizer for GCL is reasonable in my opinion. The authors demonstrate the rationality of the proposed method in both theoretical and empirical analysis.

3. The authors consider various application scenarios of the proposed method, which covers the representative downstream tasks of GCL.

Drawbacks:

1. Although the authors consider multiple application scenarios in the experimental section. In each task, the baselines are insufficient and not so representative. In the field of graph contrastive learning, some recent works should be considered as the baselines, e.g., [a - d]. These methods also consider the limitations of existing GCL framework and propose different strategies to reduce the mis-calibration issue of GCL. The authors should i) take some of them as the baselines in the experimental section and ii) introduce the recent development of GCL in the related work part, highlight the differences between these methods and the proposed method.

[a] Tong, Zekun, et al. "Directed graph contrastive learning." Advances in Neural Information Processing Systems 34 (2021): 19580-19593.

[b] Suresh, Susheel, et al. "Adversarial graph augmentation to improve graph contrastive learning." Advances in Neural Information Processing Systems 34 (2021): 15920-15933.

[c] Zhu, Yanqiao, et al. "Graph contrastive learning with adaptive augmentation." Proceedings of the Web Conference 2021. 2021.

[d] Zhao, Han, et al. "Graph Debiased Contrastive Learning with Joint Representation Clustering." IJCAI. 2021.

2. In Tables 1-4, if my understanding is correct, the proposed methods LC and ML have taken the proposed Contrast-Reg into account. However, in Table 5, the authors further introduce the methods “LC + Contrast-Reg” and “ML + Contrast-Reg”, which is confusing.

3. Besides taking GCN as the backbone, the authors should consider other backbones like GIN in the experiments, such that the usefulness of the regularizer with respect to different model architectures can be verified.

Minors:

1. The captions of Figures 1 and 2 are oversimplified. The details of the analytic experiments should be given in the captions, e.g., the dataset, the model architecture and so on.

2. Some citations are not normative. In particular, in some places, the citations should be formatted as “(Authors, Year)” rather than “Authors (Year)”. Try to use \citep instead of \cite.

3. Page 7, line 6: The notations of L_nce and L_supp^mu are inconsistent with those shown in Theorem 1.

---

> ### Author Response · Authors · 2023-06-02
> **Response to Reviewer mn1F [1/2]**
>
> ### Requested Change 1. i) take some of them as the baselines in the experimental section and ii) introduce the recent development of GCL in the related work part, highlight the differences between these methods and the proposed method.
>
> Thank you for providing the detailed information for the baselines.
>
> In the related work section, we have included all the suggested algorithms and highlighted them in red font. We want to clarify that Contrast-Reg and these algorithms are orthogonal. The latter methods aim to incorporate critical structures for improving downstream task performance, whereas Contrast-Reg aims to ensure that the performance improvements in downstream tasks and the decrease of contrastive loss are better aligned. Notably, Contrast-Reg can easily be used as a plugin into these advanced graph contrastive learning algorithms effectively.
>
> In the node classification experimental section, we have considered DiGCL (Tong et al., 2021), GCA (Zhu et al., 2021b), and GDCL (Zhao et al., 2021) as baselines. We did not include AD-GCL (Suresh et al., 2021) because our focus is on node-level tasks, whereas AD-GCL is designed for graph-level tasks. Nonetheless, we acknowledge the potential for future research to extend Contrast-Reg to graph-level tasks. To conduct our extended experiments, we utilized the official implementations and released hyperparameters of these algorithms. The experimental settings are the same as other experiments in our paper. Table 1 includes the results of GCA and GCA with Contrast-Reg, and the experiments of DiGCL and GDCL are reported in Appendix C.1. We placed DiGCL in the Appendix since it is tailored to directed graphs, while GDCL is memory inefficient and runs OOM on our machines when handling larger datasets such as PubMed.
>
> |                  | Wiki-CS        | Amazon-Photo   | Amazon-Computers | Cora           | Citeseer       | Pubmed         |
> | ---------------- | -------------- | -------------- | ---------------- | -------------- | -------------- | -------------- |
> | GCA              | 76.38±1.25     | 92.07±1.0      | 87.47±0.49       | 82.51±0.75     | 70.83±0.89     | 85.69±1.03     |
> | GCA+Contrast-Reg | **77.71±0.06** | **93.13±0.32** | **88.21±0.37**   | **83.90±1.22** | **72.39±0.88** | **87.06±0.26** |
>
>
> |                    | Cora-ml        | Citeseer       |
> | ------------------ | -------------- | -------------- |
> | DiGCL              | 76.62±1.23     | 65.54±1.39     |
> | DiGCL+Contrast-Reg | **79.74±0.62** | **68.74±1.13** |
>
>
> |                    | Cora        | Citeseer       |
> | ------------------ | -------------- | -------------- |
> | GDCL              | 85.41±0.53     | 72.98±0.73    |
> | GDCL+Contrast-Reg | **85.80±0.46** | **76.39±1.16** |
>
> We have reorganized Section 5 to highlight the general effectiveness and high compatibility of Contrast-Reg with respect to various graph contrastive learnings.

---

> ### Author Response · Authors · 2023-06-02
> **Response to Reviewer mn1F [2/2]**
>
> ### Requested Change 2. The notation in Table 1-4 is confusing.
>
> We have modified Tables 1-4 for clearer expression of the employed models.
>
> ### Requested Change 3. Response about incorporating backbones as the encoder
>
> Thank you for your suggestion of considering GIN (Xu et al., 2019) as the backbone. The experimental result is given below.
>
> | GAT backbone | Cora         | Citeseer     | Pubmed       |
> |-----------|--------------|--------------|--------------|
> | ML        | 76.90±1.20   | 70.43±0.31   | 75.05±0.51   |
> | ML+Contrast-Reg | 81.56±0.94   | 72.73±0.44   | 78.50±0.63   |
> | LC        | 74.30±1.10   | 69.30±0.42   | 72.50±0.49   |
> | LC+Contrast-Reg | 80.87±0.96   | 72.69±0.52   | 78.50±0.63   |
>
> | GIN backbone | Cora         | Citeseer     | Pubmed       |
> |-----------|--------------|--------------|--------------|
> | ML        | 77.78±0.98   | 70.78±0.68   | 80.12±0.50   |
> | ML+Contrast-Reg | 81.04±0.94   | 71.17±0.45   | 80.23±0.69   |
> | LC        | 80.23±0.61   | 71.46±0.44   | 78.46±0.51   |
> | LC+Contrast-Reg | 81.38±0.58   | 71.59±0.36   | 78.51±0.46   |
>
> We have added a new Section 6.4 in our revised draft, which discusses the generality of Contrast-Reg. We have also included the experiments that incorporate GIN and GAT (Velickovic et al., 2018) as backbones in this section. The results of these additional experiments demonstrate the generality and effectiveness of plugging Contrast-Reg into various graph contrastive learning settings that utilize different similarity definitions and GNN encoder backbones.
>
> ### Requested Change 4. Unclear description in Figure 1-2.
>
> We have updated Figures 1 and 2 with more details of the dataset and the model architecture.
>
> ### Requested Change 5. Some citations are not normative.
>
> We have fixed the citation problem in the revision.
>
> ### Requested Change 6. Inconsistent Notation.
>
> We have fixed the notation mistakes that are inconsistent with Theorem 1.
>
> [1] Zekun Tong, Yuxuan Liang, Henghui Ding, Yongxing Dai, Xinke Li, and Changhu Wang. Directed graph contrastive learning. In NeurIPS ‘21.
>
> [2] Yanqiao Zhu, Yichen Xu, Feng Yu, Qiang Liu, Shu Wu, and Liang Wang. Graph contrastive learning with adaptive augmentation. In WWW ’21.
>
> [3] Han Zhao, Xu Yang, Zhenru Wang, Erkun Yang, and Cheng Deng. Graph debiased contrastive learning with joint representation clustering. In IJCAI ‘21.
>
> [4] Susheel Suresh, Pan Li, Cong Hao, and Jennifer Neville. Adversarial graph augmentation to improve graph contrastive learning. In NeurIPS ‘21.
>
> [5] Keyulu Xu, Weihua Hu, Jure Leskovec, and Stefanie Jegelka. How powerful are graph neural networks? In ICLR ’19.
>
> [6] Petar Velickovic, Guillem Cucurull, Arantxa Casanova, Adriana Romero, Pietro Liò, and Yoshua Bengio. Graph attention networks. In ICLR ’18.

---

> ### Author Response · Authors · 2023-06-12
> **Thank you and we welcome further questions and comments**
>
> We thank the reviewer again for the constructive feedback. We hope that most of the concerns have been addressed by our response. Please let us know if there is any outstanding concern, we are very happy to follow up and discuss them.

---

### Review · Reviewer_kYUe · 2023-05-15

**Summary Of Contributions:**

The authors argue that existing graph contrastive learning methods suffer from uncertainty miscalibration. To mitigate the pathology, they proposed to add a regularization term "Contrast-Reg" to the standard NCE loss and they empirically show that it will reduce the expected calibration error (ECE) and thus improve the calibration. Moreover, they show that this strategy may improve the generalization in downstream tasks from a theoretical analysis.

**Audience:**

Yes

**Broader Impact Concerns:**

I have no concerns about the broader impacts and the ethical implications.

**Claims And Evidence:**

Yes

**Requested Changes:**

See the weaknesses above.

**Strengths And Weaknesses:**

Disclaimer: I am not familiar with graph contrastive learning, so I will comment from the perspective of uncertainty calibration.

Strengths:
1. The analysis of why ECE can be high (claim 4.1) in existing graph contrastive learning methods is interesting.
2. The authors theoretically justified why their method could potentially improve generalization in downstream tasks (Theorem 1).
3. The method is simple and the empirical results look good.

Weaknesses:
1. The method is motivated to improve the calibration during contrastive learning, however, there is no direct explanation why adding Contrast-Reg term to the NCE loss will reduce ECE (they only show it empirically).
2. ECE is a controversial calibration metric since it can only measure the population level calibration. Moreover, it is not guaranteed to be reduced with distribution shift. I am not sure whether the improvement in calibration in contrastive learning stage can be transferred to downstream tasks where there might be distribution shift (since the strategy of generating negative samples during contrastive learning stage seems to be artificial).
3. Following the above point, the reason why reducing ECE in the contrastive learning stage can even "enhance the accuracy of downstream tasks" is unclear to me. Although the authors suggest the generalization may improve by using their method (Theorem 1), it seems to be unrelated to the improved calibration in contrastive learning stage.

---

> ### Author Response · Authors · 2023-06-02
> **Response to Reviewer kYUe [1/2]**
>
> ### Requested Change 1. The method is motivated to improve the calibration during contrastive learning, however, there is no direct explanation why adding Contrast-Reg term to the NCE loss will reduce ECE (they only show it empirically).
>
> In Claim 4.1 and Appendix A.1, it is proven that the ECE value is positively correlated with $\mathbb{E}{(v,v^\prime)}[\sigma(h_v,h_{v^\prime})]$ and inversely correlated with $q^+$. Additionally, Theorem 2 in Appendix A.3 shows that Contrast-Reg reduces the variance of the learned node embeddings' norm $\Vert h\Vert$, thereby preventing the loss from decreasing due to exploding embedding norms. As a result, the norm values are lower than those obtained with the vanilla contrastive loss, leading to a reduction in $\mathbb{E}{(v,v^\prime)}[\sigma(h_v,h_{v^\prime})]$ and an increase in the representation quality. With the improvement in the representation quality, $\mathbb{E}{(v,v^\prime)}[\sigma(h_v,h_{v^\prime})]$ decreases and $q^+$ increases, thereby resulting in a decreased ECE value.
>
> Thanks for your comment and we have added a discussion about why adding the Contrast-Reg term to the NCE loss will reduce ECE in Section 4.2.
>
> ### Requested Change 2. ECE is a controversial calibration metric since it can only measure the population level calibration. Moreover, it is not guaranteed to be reduced with distribution shift. I am not sure whether the improvement in calibration in contrastive learning stage can be transferred to downstream tasks where there might be distribution shift (since the strategy of generating negative samples during contrastive learning stage seems to be artificial).
>
> We would like to clarify that, following the common setting as previous graph contrastive methods (Suresh et al., 2021; Velickovic et al., 2019; Zhu et al., 2021b; Zhao et al., 2021), **our paper focuses on the in-distribution generalization ability of GNN models**, a.k.a. inductive ability (Hamilton et al., 2017), which aims to predict the label of unseen nodes/edges/graphs within the same domain. Therefore, while ECE may not be guaranteed to be reduced with distribution shift, it is a suitable analyzer tool for quantifying whether the pseudo-labels in graph contrastive learning are aligned with the downstream tasks' labels. A smaller ECE value indicates the decrease of unsupervised loss during the training stage, which implies an improvement in the accuracy of the downstream tasks. It is important to note that ECE functions purely as an analytical tool; it is not incorporated into the training loss function. Thus, it does not risk leaking information about the labels of the downstream tasks. We also clarify that the negative sampling strategies used in our paper are common strategies employed in the contrastive learning approaches in the literature (Hamilton et al., 2017;Velickovic et al., 2019;Yang et al., 2020b).
>
> We further remark that the out-of-domain distribution shift, which tests the GNN’s extrapolation ability on unseen features/structure (Chen et al., 2022; Wu et al., 2022) (the setting of adding an additional feature dimension, or training on small graphs while predicting on larger graphs), is beyond the scope of our paper. We acknowledge that handling distribution-shift with ECE requires additional design (Wald et al., 2021), which can be an interesting future direction to extend our analysis and results to out-of-distribution settings.

---

> ### Author Response · Authors · 2023-06-02
> **Response to Reviewer kYUe [2/2]**
>
> ### Requested Change 3. Following the above point, the reason why reducing ECE in the contrastive learning stage can even "enhance the accuracy of downstream tasks" is unclear to me. Although the authors suggest the generalization may improve by using their method (Theorem 1), it seems to be unrelated to the improved calibration in contrastive learning stage.
>
> As we have discussed in our response to “Requested Change 2” above, ECE serves to quantify whether the pseudo-labels in graph contrastive learning are aligned with the downstream tasks' labels. Note that there are no downstream task’s labels but only the pseudo-labels available during the training stage. If the pseudo-labels and the groundtruth labels are aligned (by decreasing ECE), then we can claim that minimizing the unsupervised loss enables enhancing the accuracy of the downstream tasks. Therefore, reducing ECE in the contrastive learning stage can "enhance the accuracy of downstream tasks".
>
> [1] Will Hamilton, Zhitao Ying, and Jure Leskovec. Inductive representation learning on large graphs. In NeurIPS ’17.
>
> [2] Yongqiang Chen, Yonggang Zhang, Yatao Bian, Han Yang, Kaili Ma, Binghui Xie, Tongliang Liu, Bo Han, and James Cheng. Learning causally invariant representations for out-of-distribution gener- alization on graphs. In NeurIPS ‘22.
>
> [3] Qitian Wu, Hengrui Zhang, Junchi Yan, and David Wipf. Handling distribution shifts on graphs: An invariance perspective. In ICLR ‘22.
>
> [4] Yoav Wald, Amir Feder, Daniel Greenfeld, and Uri Shalit. On calibration and out-of-domain gen- eralization. In NeurIPS ‘21.
>
> [5] Petar Velickovic, William Fedus, William L. Hamilton, Pietro Liò, Yoshua Bengio, and R. Devon Hjelm. Deep graph infomax. In ICLR ’19.
>
> [6] Zhen Yang, Ming Ding, Chang Zhou, Hongxia Yang, Jingren Zhou, and Jie Tang. Understanding negative sampling in graph representation learning. In KDD ’20.
>
> [7] Yanqiao Zhu, Yichen Xu, Feng Yu, Qiang Liu, Shu Wu, and Liang Wang. Graph contrastive learning with adaptive augmentation. In WWW ’21.
>
> [8] Han Zhao, Xu Yang, Zhenru Wang, Erkun Yang, and Cheng Deng. Graph debiased contrastive learning with joint representation clustering. In IJCAI ‘21.
>
> [9] Susheel Suresh, Pan Li, Cong Hao, and Jennifer Neville. Adversarial graph augmentation to improve graph contrastive learning. In NeurIPS ‘21.

---

> ### Author Response · Authors · 2023-06-12
> **Thank you and we welcome further questions and comments**
>
> We thank the reviewer again for the constructive feedback. We hope that most of the concerns have been addressed by our response. Please let us know if there is any outstanding concern, we are very happy to follow up and discuss them.

---

### Review · Reviewer_gyCv · 2023-05-20

**Summary Of Contributions:**

This paper focuses on better graph contrastive learning.  This paper presents a regularization method called Contrast-Reg to improve the performance of graph contrastive learning algorithms and graph neural network models. It addresses calibration issues by maintaining high-quality node representations that improve accuracy in downstream tasks, avoiding overfitting to spurious features. Both theoretical and empirical evidence support the effectiveness of Contrast-Reg across various tasks.

**Audience:**

Yes

**Claims And Evidence:**

Yes

**Requested Changes:**

1. Correct Citations: Ensure only relevant citations are in the correct format.
2. Clarify introduction: Clarify the meaning of the quality of embeddings in the introduction.
3. Clarify the Motivation: Provide evidence for the claim in the abstract about inconsistent effectiveness.
4. Improve readability by clarifying the motivation and influence of ECE.
5. Experiments: Include more recent baselines for node classification and integrate link prediction methods.


**Strengths And Weaknesses:**

**Strengths**
1. The problem shown in Figure 1 seems to show that the NCEloss-based graph contrastive learning is problematic in some aspects.
2. The proposed method is easy to implement and plugged into the current method.
3. The provided Theoretical analysis seems sound to me.


**Weaknesses**
1. The citation of this paper is super random. Just cite the papers that relate to wide-range terms. such as in the Introduction (page 1), “graph representation learning algorithms Velickovic et al. (2019); Peng et al. (2020); Tang et al. (2015); Perozzi et al. (2014); Grover & Leskovec (2016); Ahmed et al. (2013); Cao et al. (2015); Qiu et al. (2018); Chen et al. (2018); Kipf & Welling (2017); Hamilton et al. (2017); Velickovic et al. (2018); Xu et al. (2019); Qu et al. (2019) have been proposed”. Such a way for citation is not an accurate and reasonable way for citation. (as well the citation is not in the correct formation, please use the \citet{} and \citep{} correctly.)
2. The presentation is not clear. In the introduction, what is the meaning of “the quality of the embeddings generated by the model, $\delta(h\_v, h\_{v’})$”? And how this influences the learned embedding? Please try to make the introduction self-contained.
3. The motivation is not clear. This paper claims that “to new tasks or data has proven to be inconsistently effective” (in the abstract), However, there is no analysis or experimental evidence to support this point.
4. This paper overall a bit hard to follow, mainly because of the unclear motivation, and unclear meaning/influence of ECE.
5. The experiments can be improved.
   - For node classification, only DGI and GMI, which are not new, are referenced as the most recent or relevant baselines. There are numerous unsupervised graph learning methods available. Furthermore, the performance is subpar. Some models cannot outperform the baseline, while others show only marginal improvements.
   -  For link prediction, this paper neglects to consider link prediction methods.

---

> ### Author Response · Authors · 2023-06-02
> **Response to Reviewer gyCv [1/2]**
>
> ### Requested Change 1. Correct Citations: Ensure only relevant citations are in the correct format.
>
> Thank you for your careful review of our paper. We have addressed the problem of the citation format and made changes to the citations in the introduction section.
>
> ### Requested Change 2. Clarify introduction: Clarify the meaning of the quality of embeddings in the introduction.
>
> In the previous version of the paper, the quality of the embeddings generated by the model, represented as $\sigma(h_v\cdot h_{v'})$, refers to the predicted probability that node $v^\prime$ shares the same label with node $v$. Here, $h_v$ and $h_{v^\prime}$ are the embeddings of nodes $v$ and $v^\prime$, respectively. We understand that this sentence was confusing and have rewritten it as follows for clarity: "To address this issue, we first adapt the expected calibration error (ECE) to the graph contrastive learning framework to assess the discrepancy between the predicted probability and the ground truth for the contrastive pairs."
>
> ### Requested Change 3. Clarify the Motivation: Provide evidence for the claim in the abstract about inconsistent effectiveness.
>
> We understand the description “inconsistently effective” is rather confusing and inappropriately used and therefore we have rewritten the original sentence. We first explain what we actually mean by “inconsistently effective” below, before we present the new sentence. The description “directly applying contrastive pairs and graph neural network (GNN) models to new tasks or data has proven to be inconsistently effective.” refers to the experimental results reported in Table 5. For example, LC, which is commonly used to incorporate community structure, such as in AND (Huang et al., 2019) and GDCL (Zhao et al., 2021), achieves the state-of-the-art results on the Reddit social network but underperforms on the Wikipedia network (measured by the classification accuracy). Our objective, therefore, is to investigate the cause of **this inconsistency in the algorithm's performance**, particularly considering that both datasets have community structures. Therefore, we have now rewritten the sentence in the abstract as: "However, in unsupervised graph contrastive learning, some contrastive pairs can contradict the truths in downstream tasks, so that the decrease of losses on these pairs undesirably harms the performance in the downstream tasks. To assess the discrepancy between the prediction and the groundtruth in the downstream tasks for these contrastive pairs, we adopt the expected calibration error (ECE) to the graph contrastive learning framework."
>
> |                              | Wiki                  |     Reddit         |
> | ---------------           | --------------        | --------------       |
> | LC                        | 65.14±1.48       |    94.42±0.03   |
> | LC+Contrast-Reg | **69.19±1.13** | **94.43±0.03**|

---

> ### Author Response · Authors · 2023-06-02
> **Response to Reviewer gyCv [2/2]**
>
> ### Requested Change 4. Improve readability by clarifying the motivation and influence of ECE.
>
> Following the above response, the motivation of imposing ECE is to calibrate the pseudo-labels used during the training stage and the ground truth labels in the downstream tasks, so as to get insights on how to ensure that minimizing the graph contrastive loss results in better performance of the downstream tasks.
>
> ECE is not directly incorporated into the model or the training loss. Instead, we use ECE to identify the key factors that influence the discrepancy between the prediction in unsupervised training and the accuracy in downstream tasks (Claim 4.1). The key factors motivate us to design a plugin regularizer, Contrast-Reg, upon different advanced contrastive loss designs. Section 4.2 shows that imposing Contrast-Reg leads to the decrease of the ECE value.
>
> ### Requested Change 5. Experiments: Include more recent baselines for node classification and integrate link prediction methods.
>
> We appreciate your suggestion of including more recent baselines for node classification and link prediction tasks.
>
> For the node classification task, we have included GCA (Zhu et al., 2021b), DiGCL (Tong et al., 2021), and GDCL (Zhao et al., 2021) as the baselines, which are advanced graph contrastive learning algorithms. Table 1 reports the results of GCA on various datasets. By incorporating Contrast-Reg, the node classification accuracy of GCA is further improved. This validates that Contrast-Reg enhances GCA's generalization ability that brings improvements in downstream tasks by improving the quality of the learned embeddings. Similar results using DiGCL and GDCL can be found in Appendix C.1.
>
> |                  | Wiki-CS        | Amazon-Photo   | Amazon-Computers | Cora           | Citeseer       | Pubmed         |
> | ---------------- | -------------- | -------------- | ---------------- | -------------- | -------------- | -------------- |
> | GCA              | 76.38±1.25     | 92.07±1.0      | 87.47±0.49       | 82.51±0.75     | 70.83±0.89     | 85.69±1.03     |
> | GCA+Contrast-Reg | **77.71±0.06** | **93.13±0.32** | **88.21±0.37**   | **83.90±1.22** | **72.39±0.88** | **87.06±0.26** |
>
>
> |                    | Cora-ml        | Citeseer       |
> | ------------------ | -------------- | -------------- |
> | DiGCL              | 76.62±1.23     | 65.54±1.39     |
> | DiGCL+Contrast-Reg | **79.74±0.62** | **68.74±1.13** |
>
>
> |                    | Cora        | Citeseer       |
> | ------------------ | -------------- | -------------- |
> | GDCL              | 85.41±0.53     | 72.98±0.73    |
> | GDCL+Contrast-Reg | **85.80±0.46** | **76.39±1.16** |
>
> For the link prediction task, we have integrated Contrast-Reg into two commonly used contrastive link prediction baselines: ML-GCN (Shiao et al., 2022) and GRACE (Zhu et al., 2020). Table 3 shows that Contrast-Reg improves the prediction accuracy of both models across five datasets. This validates the generality of Contrast-Reg as a plugin in contrastive link prediction tasks, enhancing the performance of various baselines.
>
> |                    | Cora        | Citeseer       | Pubmed |  Wiki |
> | ------------------ | -------------- | -------------- |-------------- | -------------- |
> | GRACE              | 92.82±0.65 |93.14±0.73|96.05±0.91|87.72±1.16|
> | GRACE+Contrast-Reg |**93.60±0.57** |**93.24±0.62**|**96.60±1.56**|**84.90±1.84**|
> |ML-GCN|94.81±0.55 |96.38±0.63|92.86±0.26|89.84±0.69|
> |ML-GCN+Contrast-Reg|**95.18±0.31**|**96.83±0.55**|**95.10±0.22**|**92.70±0.56**|
>
>
> [1] Jiabo Huang, Qi Dong, Shaogang Gong, and Xiatian Zhu. Unsupervised deep learning by neighbourhood discovery. In ICML ’19.
>
> [2] Han Zhao, Xu Yang, Zhenru Wang, Erkun Yang, and Cheng Deng. Graph debiased contrastive learning with joint representation clustering, In IJCAI ‘21.
>
> [3] Yanqiao Zhu, Yichen Xu, Feng Yu, Qiang Liu, Shu Wu, and Liang Wang. Graph contrastive learning with adaptive augmentation. In WWW ’21.
>
> [4] Zekun Tong, Yuxuan Liang, Henghui Ding, Yongxing Dai, Xinke Li, and Changhu Wang. Directed graph contrastive learning. In NeurIPS ‘21
>
> [5] William Shiao, Zhichun Guo, Tong Zhao, Evangelos E. Papalexakis, Yozen Liu, and Neil Shah. Link predic- tion with non-contrastive learning. In abs/2211.14394, 2022.
>
> [6] Yanqiao Zhu, Yichen Xu, Feng Yu, Qiang Liu, Shu Wu, and Liang Wang. Deep graph contrastive representation learning. In ICML GRL ‘20.

---

> ### Author Response · Authors · 2023-06-12
> **Thank you and we welcome further questions and comments**
>
> We thank the reviewer again for the constructive feedback. We hope that most of the concerns have been addressed by our response. Please let us know if there is any outstanding concern, we are very happy to follow up and discuss them.

---

### Author Response · Authors · 2023-06-02
**We have uploaded a revised version [Updated on 03 Jun.]**

Dear reviewers,

Thank you for your efforts in reviewing our paper and your many constructive comments that have helped improve our paper. We have carefully revised our paper following your comments and suggestions. We highlight the revisions in red fonts in the revised paper and the new experimental results with yellow background in the tables in Section 6. A summary of the revisions are listed as follows:

**About presentation**

- We have addressed the issue of the citation format and made necessary modifications to the citations in the introduction section.
- We have revised the abstract and introduction to provide a clearer explanation of the motivation and influence of ECE.
- We have improved the presentation of Tables 1-4 for better expression of the models used, the captions of Figures 1-2 for more detailed experimental settings, and corrected the notational errors.
- We have added the explanation for why adding the Contrast-Reg term to the NCE loss will reduce ECE in Section 4.1.

**About experiments**

- Section 5: We have restructured Section 5 to highlight the high compatibility of Contrast-Reg with various graph contrastive learning techniques.
- Section 6.2.1: We have added GCA as a baseline and obtained improved results on GCA with Contrast-Reg. We have also updated the experimental descriptions.
- Section 6.2.3: We have included GRACE and ML-GCN obtained improved results on GRACE and ML-GCN with Contrast-Reg. We have also updated the experimental descriptions.
- Section 6.4: We have conducted experiments across different encoder backbones, which demonstrate the efficacy of Contrast-Reg as a regularization term for graph contrastive learning algorithms that use different similarity definitions and GNN encoder backbones.
- Appendix C.1: We plugged Contrast-Reg into DiGCL and GDC, and demonstrated that Contrast-Reg serves as a seamless plugin to these advanced graph contrastive learning algorithms, even when applied to directed graphs (DiGCL).
- Section 2: We have updated Section 2 to include all the newly added baselines in related works and clearly differentiate Contrast-Reg from them.

In addition, we have provided a link of our code for reproducing the results in our paper: [code link](https://anonymous.4open.science/status/Contrast-Reg).

Below, we further give our responses to each reviewer’s specific questions. Please let us know if you have any further questions.

---

### Decision · Action_Editors · 2023-06-27

**Recommendation:** Accept with minor revision

**Comment:**

Reviewers agreed that the problem of calibration in graph contrastive learning is under-explored so they welcomed this study.

In initial review stage, reviewers mainly raised two concerns:
1. Explanations on why ECE is related to contrastive learning performance, and how would the proposed regulariser help.
2. Benchmarks selected in the experiment section are outdated.

In revision the authors clarified their theory and added a few more experiments. The questions related to theory are largely addressed, but some reviewers are still unhappy about the selected baselines. In particular they suggested the following baselines to be considered:

- S2GAE: Self-Supervised Graph Autoencoders are Generalizable Learners with Graph Masking, WSDM2023

- GraphMAE2: A Decoding-Enhanced Masked Self-Supervised Graph Learner, TheWebConf2023

- SeeGera: Self-supervised Semi-implicit Graph Variational Auto-encoders with Masking, TheWebConf2023

- GraphMAE: Self-Supervised Masked Graph Autoencoders, KDD2022

From my point of view the 2023 papers can be viewed as concurrent works so it is OK not to consider them. But I agree that given the fast evolution of the graph NN field, it would make sense to include the latest (2022) graph contrastive learning methods (e.g., the last one above) in comparison.

Given that the paper's contributions are not solely based on empirical improvements, and the reviewers are welcoming the theory contributions, I recommend acceptance but conditioned on including one of the latest graph contrastive learning methods as baselines.

**Audience:**

Researchers working on the following topics: graph data, graph neural network, contrastive learning, uncertainty quantification.

**Claims And Evidence:**

This paper considers uncertainty estimation for the process of graph contrastive learning. In particular, the authors constructed the ECE metric on the graph contrastive learning task (i.e., predicting whether two presented nodes are from positive pairs or negative pairs), and proposed Contrast-Reg as a regulariser that aims to reduce the constructed ECE metric.

The authors conducted a few theoretical analyses demonstrating the positive correlation between the proposed regulariser and contrastive learning performance, and between the proposed regulariser and ECE.

Empirical results on node classification, link prediction and graph clustering showed improved performance, although not all of the improvements are significant.

---

> ### Author Response · Authors · 2023-07-02
> **Thank you for reviewing our paper**
>
> We would like to thank the AE and the anonymous reviewers for their constructive suggestions.
>
> We have revised our paper according to the reviewers' suggestions. In particular, we included "GraphMAE: Self-Supervised Masked Graph Autoencoders, KDD2022" as a comparison in the experiments (Sectoin 6.2.1), as suggested by the AE and reviewers. In addition, we have also discussed the four related works mentioned by the AE in Section 2.